# Quartet of Diffusions: Structure-Aware Point Cloud Generation through Part and Symmetry Guidance

**Chenliang Zhou, Fangcheng Zhong,** *Weihao Xia, Albert Miao, Canberk Baykal, Cengiz Oztireli**
Department of Computer Science and Technology, University of Cambridge

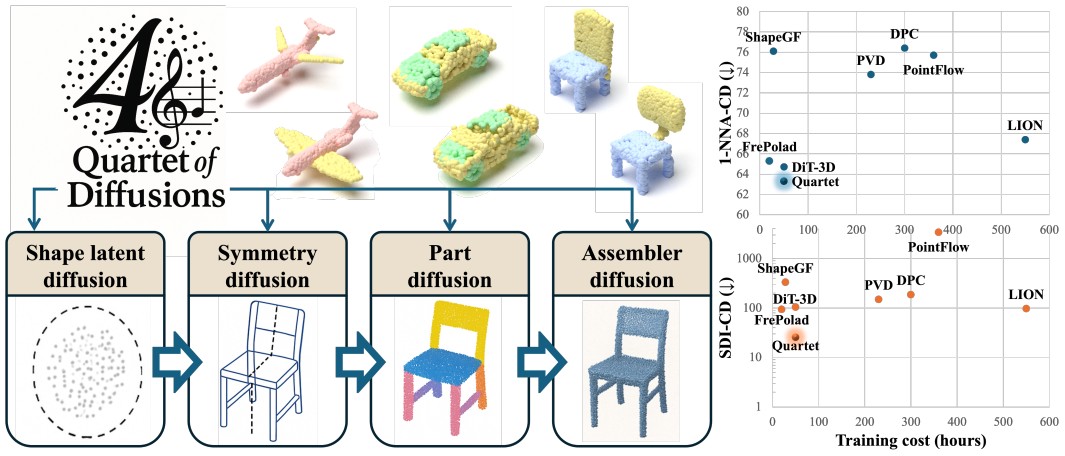

Figure 1: Our method, the Quartet of Diffusions, harmoniously orchestrates an interpretable and controllable pipeline for structure-aware, high-quality, diverse, and efficient point cloud generation guided by part and symmetry clues.

## Abstract

We introduce the *Quartet of Diffusions*, a structure-aware point cloud generation framework that explicitly models part composition and symmetry. Unlike prior methods that treat shape generation as a holistic process or only support part composition, our approach leverages four coordinated diffusion models to learn distributions of global shape latents, symmetries, semantic parts, and their spatial assembly. This structured pipeline ensures guaranteed symmetry, coherent part placement, and diverse, high-quality outputs. By disentangling the generative process into interpretable components, our method supports fine-grained control over shape attributes, enabling targeted manipulation of individual parts while preserving global structural consistency. A central global latent further reinforces structural coherence across assembled parts. Our experiments show that the Quartet achieves state-of-the-art performance in generating high-quality and diverse point clouds while maintaining symmetry. To our best knowledge, this is the first 3D point cloud generation framework that fully integrates and enforces both symmetry and part priors throughout the generative process.

## 1 Introduction

Structure-aware 3D shape generation aims to model not only the holistic surface geometry of objects but also the **underlying organizational principles** that govern their composition and formation (Chaudhuri et al., 2020). A truly structure-aware representation must comprise two essential components (Chaudhuri et al., 2020): the geometry of atomic structural elements (*e.g.*, low-level

---

*Corresponding author

parts) and the structural patterns that dictate how these elements are assembled into coherent shapes or scenes. These patterns may manifest as hierarchical decompositions, part-whole relationships, spatial composition graphs, or local and global symmetries.

In this work, we focus on 3D point cloud generation with **explicit awareness of two fundamental structural priors** commonly observed in natural and man-made objects: **part awareness** and **symmetry awareness**. *Part awareness* captures the notion that objects are composed of semantically meaningful, interrelated components with geometrically distinct properties. *Symmetry awareness*, in contrast, models the intrinsic geometric regularities—such as reflective or rotational symmetries—that often govern the spatial arrangement of parts. Importantly, part awareness serves as a *prerequisite* for identifying and enforcing local symmetries, which typically operate at the level of inter-part relationships. Without symmetry, shapes risk appearing disjoint or unnatural, undermining their functional performance in practical applications (*e.g.* some prior work in Fig. 4).

Compared to unstructured 3D data, structured representations offer substantial advantages. They not only enhance the *visual plausibility* and *structural coherence* of generated shapes, but also facilitate downstream tasks such as part segmentation, alignment, structure inference, and fine-grained shape editing (Schor et al., 2019; Li et al., 2020; Hertz et al., 2022; Koo et al., 2023). However, existing generative approaches largely remain structure-oblivious – treating point cloud synthesis as an unconstrained distribution learning problem. These methods often prioritize global visual fidelity and geometry-level details in the distribution, while neglecting the explicit modeling of inherent structure (Vahdat et al., 2022; Yang et al., 2019; Mo et al., 2023; Zhou et al., 2021a). As a result, they tend to produce shapes that suffer from poor part organization, broken or inconsistent symmetries, and limited generalization to unseen or complex object categories (Li et al., 2020). While a few recent efforts support part-level generation, they rarely incorporate symmetry as a *learned*, *relational prior*. To date, there remains a notable gap: the absence of unified, principled frameworks that jointly encode *part-level composition* and *symmetry structures* to guide the generation of coherent, interpretable, and generalizable 3D shapes.

To address this, we propose the *Quartet of Diffusions* (the *Quartet*), a structure-aware point cloud generation framework that explicitly leverages part and symmetry guidance. Specifically, our pipeline employs four diffusion models (Ho et al., 2020) to learn the distributions of shape latents, symmetries, semantic parts, and assemblers that assemble the full point cloud from parts. By explicitly modeling these structural distributions, the ensemble of four diffusions harmoniously orchestrates an effective pipeline for structure-aware point cloud generation through part and symmetry guidance: Our method enables the generation of high-quality, diverse point clouds with guaranteed symmetry. The disentangled structural representation makes the generation process interpretable and controllable, facilitating targeted modifications to individual parts while preserving global consistency. Structural coherence is further reinforced by a global shape latent, which anchors part assembly to the overall geometry. This approach addresses key limitations in structure modeling and represents the first 3D shape generation method to guarantee symmetry in the generated shapes.

## 2 RELATED WORK

**Symmetry in 3D shapes** Symmetry is a fundamental geometric property observed across natural and human-made objects. Extensive research exists on symmetry detection (Mitra et al., 2013a; Atallah, 1985; Illingworth & Kittler, 1988; Mitra et al., 2006; Je et al., 2024; Zhou et al., 2021b; Gao et al., 2020; Fukunaga & Hostetler, 1975; Comaniciu & Meer, 2002) (for more details, see Appendices A and B.1). Symmetry clues are widely used in various 3D vision tasks including acquisition and representation (Yang et al., 2024; Buades et al., 2008; Li et al., 2010; Pauly et al., 2005; Thrun & Wegbreit, 2005; Xu et al., 2009; Zheng et al., 2010), classification (Kazhdan et al., 2004; Martinet et al., 2006; Podolak et al., 2006), perception (Reisfeld et al., 1995) , manipulation (Mitra et al., 2007; Podolak et al., 2007; Panozzo et al., 2012; Gal et al., 2009; Mehra et al., 2009), reconstruction (Phillips et al., 2016; Xu et al., 2024; Tulsiani et al., 2020), inverse rendering (Wu et al., 2020b), and refinement (Mitra et al., 2007). However, symmetry-aware 3D shape generation remains underexplored. Aside from our method, few approaches guarantee symmetry. PAGENet (Li et al., 2020) promotes symmetry via an MSE loss between a shape and its reflection, but it is limited to reflectional symmetry and does not guarantee it.

**Part-based 3D shape generation** Part-based 3D shape generation leverages the modular structure of objects (Mitra et al., 2014; 2013b) and emerges as a vital paradigm for modeling complex geometries (Zerroug & Nevatia, 1999; Kim et al., 2013), enhancing diversity (Schor et al., 2019; Chen et al., 2024) and facilitating downstream tasks including recognition (Hoffman & Richards, 1984), retrieval (Chang et al., 2015; Mitra et al., 2014), and manipulation (Huang et al., 2014). Early methods focus on geometric template fitting or hierarchical part assembly (Funkhouser et al., 2004; Bokeloh et al., 2010; Kalogerakis et al., 2012; Berthelot et al., 2017; Cohen-Or & Zhang, 2016; Fish et al., 2014). More recent work harnesses implicit shape representations (Koo et al., 2023; Hertz et al., 2022; Talabot et al., 2025; Hui et al., 2022; Genova et al., 2019; 2020). For example, PartSDF (Talabot et al., 2025) models parts using implicit neural fields, enabling both continuous interpolation and discrete composition.

Deep neural networks are widely adopted for part-based shape modeling (Schor et al., 2019; Li et al., 2020; Dubrovina et al., 2019; Li et al., 2024; Wu et al., 2020a; Huang et al., 2015; Li et al., 2017; Mo et al., 2019; Zou et al., 2017; Nash & Williams, 2017; Wu et al., 2019; Wang et al., 2018; 2019; Gao et al., 2019). For instance, CompoNet (Schor et al., 2019) enhances diversity by varying both parts and their compositions, while PASTA (Li et al., 2024) generates shapes conditioned on part arrangement for fine-grained control. Similarly, our Quartet performs structure-aware 3D generation by modeling both parts and their compositions, with the added benefit of symmetry enforcement.

**Diffusion models** Diffusion models (Ho et al., 2020) are generative models based on Markovian diffusion processes (see Appendix B.3 for more details). They have demonstrated remarkable performance in various domains (Croitoru et al., 2023; Cao et al., 2024; Yang et al., 2023), particularly in image (Dhariwal & Nichol, 2021; Ho et al., 2020; Rombach et al., 2022; Ramesh et al., 2022; Brooks et al., 2023; Saharia et al., 2022), speech (Chen et al., 2020; Jeong et al., 2021; Liu et al., 2022), video (Ho et al., 2022; Xing et al., 2024; Luo et al., 2023; Yang et al., 2023), 3D scene (Wei et al., 2023; Zhai et al., 2023; Tang et al., 2024), and 3D object generation (Zhou et al., 2021a; Luo & Hu, 2021; Vahdat et al., 2022; Nakayama et al., 2023; Wu et al., 2023; Mo et al., 2023; Liu et al., 2019; Zhou et al., 2024; Koo et al., 2023). Accordingly, our Quartet recruits four diffusion models to learn the distributions of shape latents, symmetries, parts, and assemblers.

## 3 METHOD

We aim to model the distribution of point clouds $\mathbf{x} \in \mathcal{X} \subseteq \mathbb{R}^{3 \times N}$ as that over a collection of semantic parts $\{\mathbf{p}_j\}_{j=1}^M$ through part and symmetry guidance. Specifically, we view each point cloud $\mathbf{x} = \bigcup_{j=1}^M T_j \mathbf{p}_j$ as a composition of its $M$ semantic parts and their corresponding assemblers. Each assembler $T_j$ is a transformation comprising translation, rotation, and scaling in 3D Euclidean space, mapping the part $\mathbf{p}_j$ to the correct position, orientation, and scale in the original point cloud.

The Quartet models four distributions using parameterized diffusion models: 1. point cloud shape latents $p_{\boldsymbol{\theta}}(\mathbf{z})$ (Sec. 3.1), 2. part-wise symmetries $p_{\boldsymbol{\zeta}}(\mathcal{S}_j \mid \mathbf{z})$ (Sec. 3.2), 3. point cloud parts $p_{\boldsymbol{\xi}}(\mathbf{p}_j \mid \mathcal{S}_j, \mathbf{z})$ (Sec. 3.3), and 4. assemblers $p_{\boldsymbol{\psi}}(T_j \mid \mathbf{w}_j, \mathbf{p}_j, \mathcal{S}_j, \mathbf{z})$, additionally conditioned on part latent $\mathbf{w}_j$ (Sec. 3.4). These components collectively define the *Quartet of Diffusions* architecture. Pipeline overviews are shown in Figs. 1 and 2.

Modeling point cloud distributions in this way offers several key advantages: 1. It introduces variability at both the shape and assembly levels, enabling combinatorially greater diversity; 2. By explicitly generating and enforcing symmetry early in the pipeline, the model produces outputs with more realistic and consistent symmetric properties; 3. The separate modeling of symmetries, parts, and assembly allows for better interpretability and fine-grained control – individual parts can be manipulated independently without compromising global structure; and 4. A central shape latent $\mathbf{z}$ ensures structural coherence across all components. These benefits are demonstrated through extensive experiments in Sec. 4.

The Quartet is trained by sequentially optimizing the parameters of four distributions to fit the given dataset $\mathcal{X}$. Point cloud generation is performed in two phases: part generation and part assembly. In part generation, a shape latent $\mathbf{z}$ is first sampled from the latent distribution $p_{\boldsymbol{\theta}}(\mathbf{z})$. Conditioned on $\mathbf{z}$, symmetry groups $\mathcal{S}_j$ are sampled from $p_{\boldsymbol{\zeta}}(\mathcal{S}_j \mid \mathbf{z})$, and parts $\mathbf{p}_j$ are drawn from $p_{\boldsymbol{\xi}}(\mathbf{p}_j \mid \mathcal{S}_j, \mathbf{z})$ for $j = 1, 2, \dots, M$; In part assembly, parts $\mathbf{p}_j$ are encoded into latent representa-

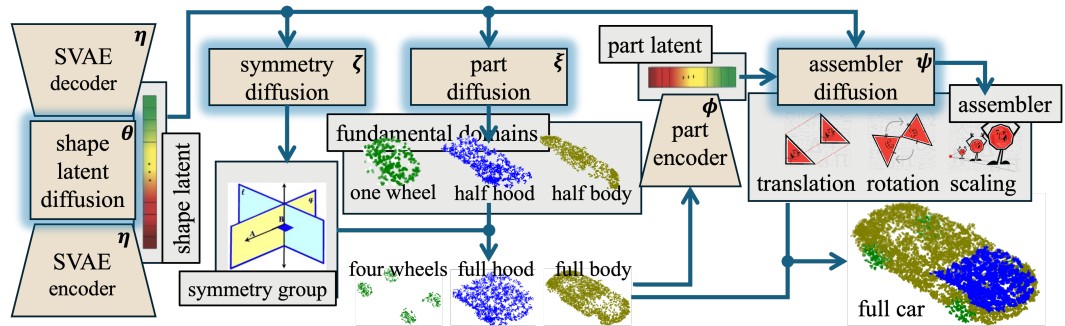

Figure 2: Overview of the Quartet's architecture. Four diffusions are employed to learn the distributions of shape latents, symmetries, parts, and assemblers. By explicitly modeling different distributions, the Quartet provides an interpretable and controllable framework to generate high-quality, diverse 3D shapes with guaranteed symmetry. Beige blocks denote learnable modules; gray blocks indicate outputs directly generated from the models.

tions $\mathbf{w}_j$ using an encoder $q_\phi\left(\mathbf{w}_j \mid \mathbf{p}_j, \mathbf{z}\right)$, and the corresponding assemblers $T_j$ are sampled from $p_\psi\left(T_j \mid \mathbf{w}_j, \mathbf{p}_j, \mathcal{S}_j, \mathbf{z}\right)$. These assemblers are then applied to the parts to reconstruct the full point cloud

$$\tilde{\mathbf{x}} := \bigcup_{j=0}^{M} T_j \mathbf{p}_j. \tag{1}$$

Mathematically, the generation process can thus be formulated as

$$p\left(\left\{\left(\mathbf{p}_j, T_j, \mathcal{S}_j, \mathbf{w}_j\right)\right\}_{j=0}^{M}, \mathbf{z}\right) = p_\theta\left(\mathbf{z}\right) p\left(\left\{\left(\mathbf{p}_j, T_j, \mathcal{S}_j, \mathbf{w}_j\right)\right\}_{j=0}^{M} \Big| \mathbf{z}\right) = p_\theta\left(\mathbf{z}\right) \prod_{j=0}^{M} p(\mathbf{p}_j, T_j, \mathbf{w}_j \mid \mathbf{z}) \tag{2}$$

$$= p_\theta\left(\mathbf{z}\right) \prod_{j=0}^{M} p_\zeta\left(\mathcal{S}_j \mid \mathbf{z}\right) p_\xi\left(\mathbf{p}_j \mid \mathcal{S}_j, \mathbf{z}\right) q_\phi\left(\mathbf{w}_j \mid \mathbf{p}_j, \mathbf{z}\right) p_\psi\left(T_j \mid \mathbf{w}_j, \mathbf{p}_j, \mathcal{S}_j, \mathbf{z}\right), \tag{3}$$

where the second equality in (2) follows from the conditional independence of parts given the shape latent $\mathbf{z}$. Therefore, the overall point cloud distribution can be modeled as

$$p\left(\mathbf{x}\right) = p\left(\left\{\left(\mathbf{p}_j, T_j\right)\right\}_{j=0}^{M}\right) = \int \cdots \int p\left(\left\{\left(\mathbf{p}_j, T_j, \mathcal{S}_j, \mathbf{w}_j\right)\right\}_{j=0}^{M}, \mathbf{z}\right) \left(\prod_{j=1}^{M} \mathrm{d}\mathcal{S}_j\right) \left(\prod_{j=1}^{M} \mathrm{d}\mathbf{w}_j\right) \mathrm{d}\mathbf{z} \tag{4}$$

$$\stackrel{(3)}{=} \int p_\theta\left(\mathbf{z}\right) \prod_{j=0}^{M} \left(\int \int p_\xi\left(\mathbf{p}_j \mid \mathcal{S}_j, \mathbf{z}\right) q_\phi\left(\mathbf{w}_j \mid \mathbf{p}_j, \mathbf{z}\right) p_\psi\left(T_j \mid \mathbf{w}_j, \mathbf{p}_j, \mathcal{S}_j, \mathbf{z}\right) \mathrm{d}\mathcal{S}_j \mathrm{d}\mathbf{w}_j\right) \mathrm{d}\mathbf{z}. \tag{5}$$

## 3.1 SHAPE LATENT DIFFUSION

Although we model each distribution separately, they must remain aware of the underlying point cloud to coordinate coherently. Directly conditioning on the full point cloud is computationally expensive due to its high dimensionality. Instead, we condition them on shape latents obtained from a variational autoencoder (VAE) (Kingma & Welling, 2013; Zhou et al., 2024; Vahdat et al., 2022) (see Appendix B.2 for background).

To effectively encode point clouds while preserving semantic information, we propose a novel *sparse variational autoencoder* (*SVAE*) with a latent diffusion modeling its latent distribution. SVAE builds on a VAE architecture implemented with point-voxel convolutional neural networks (PVCNNs) (Liu et al., 2019; Zhou et al., 2021a), drawing inspiration from sparse autoencoders (Ng et al., 2011). Prior work suggests that enforcing sparsity in activation layers improves interpretability (Cunningham et al., 2023; Makelov et al., 2024; Marks et al., 2024) and semantic disentanglement (Bricken et al., 2023; O'Neill et al., 2024), which is critical for our downstream tasks. To this end, we introduce a sparsity constraint on the final activation layer $a_\eta(\mathbf{x})$ of the encoder $q_\eta\left(\mathbf{z} \mid \mathbf{x}\right)$ while maximizing the VAE evidence lower bound (ELBO) $\mathcal{L}_{\text{ELBO}}$ (Eq. (24) in Appendix B.2):

$$\max_{\boldsymbol{\eta}} \mathbb{E}_{\mathbf{x}}\left[\mathcal{L}_{\text{ELBO}}(\boldsymbol{\eta}; \mathbf{x})\right] \text{ subject to } \mathbb{E}_{\mathbf{x}}\left[\|a_{\boldsymbol{\eta}}(\mathbf{x})\|_1\right] < \delta, \tag{6}$$

where $\delta > 0$ controls the sparsity strength. Leveraging Karush–Kuhn–Tucker (KKT) methods (Kuhn & Tucker, 1951; Karush, 1939), we reformulate the constrained optimization problem into a Lagrangian $\mathcal{F}(\boldsymbol{\eta}, \lambda; \mathbf{x})$ whose optimal point is a global maximum over the domain of $\boldsymbol{\eta}$ and obtain its lower bound:

$$\mathcal{F}(\boldsymbol{\eta}, \lambda; \mathbf{x}) := \mathcal{L}_{\text{ELBO}}(\boldsymbol{\eta}; \mathbf{x}) - \lambda \left( \|a_{\boldsymbol{\eta}}(\mathbf{x})\|_1 - \delta \right) \tag{7}$$

$$\geq \mathcal{L}_{\text{ELBO}}(\boldsymbol{\eta}; \mathbf{x}) - \lambda \|a_{\boldsymbol{\eta}}(\mathbf{x})\|_1 =: \mathcal{L}_{\text{SVAE}}(\boldsymbol{\eta}; \mathbf{x}), \tag{8}$$

where $\lambda$ is the KKT multiplier. The SVAE is trained by maximizing this lower bound $\mathcal{L}_{\text{SVAE}}$. The hyperparameter $\lambda$ controls the trade-off between reconstruction quality and sparsity, with $\lambda = 0$ recovering the standard ELBO.

While a simple Gaussian prior is commonly used for VAE latent distributions, evidence suggests that such a restricted prior cannot accurately capture complex latent distributions (*i.e.* the *prior hole problem* (Vahdat et al., 2021; Zhou et al., 2023)) and can degrade VAE performance (Chen et al., 2016). To address this, we follow prior work (Zhou et al., 2024) and employ a diffusion model $p_{\boldsymbol{\theta}}(\mathbf{z})$, implemented with a U-Net backbone, to learn the latent distribution. Once trained, the shape latents can be directly sampled from it. This design enables effective and interpretable feature extraction via SVAE while enhancing latent expressiveness and maintaining runtime efficiency.

## 3.2 SYMMETRY ENFORCEMENT

The Quartet guarantees symmetry by explicitly learning its distribution. Once learned, the model can generate and enforce appropriate symmetries during point cloud generation. For a 3D shape $\mathbf{p}$, a finite group of rigid transformations $\mathcal{S} = \langle \mathcal{T} \rangle \subseteq \mathrm{E}(3)$, generated by $\mathcal{T} = \{S_1, S_2, \ldots, S_n\}$, is said to be a *symmetry group* over a subset $\mathbf{d} \subseteq \mathbf{p}$ if the full shape $\mathbf{p}$ can be reconstructed by $\mathcal{S}\mathbf{d}$, the application of $\mathcal{S}$ on $\mathbf{d}$, which is defined to be the union of all images under the transformations in $\mathcal{S}$:

$$\underbrace{\bigcup_{S \in \mathcal{S}} S\mathbf{d}}_{\mathcal{S}\mathbf{d}} = \mathbf{p}, \tag{9}$$

where the group generation is defined as

$$\langle \mathcal{T} \rangle := \left\{ \prod_{i=1}^{n} S_i \,\middle|\, S_i \in \mathcal{T}, n \in \mathbb{N} \right\}. \tag{10}$$

The minimal such subset $\mathbf{d}$ is called the *fundamental domain* for $\mathcal{S}$. Figure 3 illustrates fundamental domains for various parts in point cloud airplanes, cars, and chairs.

As translational symmetry implies an infinite shape, for tractability we restrict symmetry groups to those generated by at most two transformations – reflections or rotations with angles $\alpha \geq \frac{\pi}{18}$ such that $\frac{2\pi}{\alpha} \in \mathbb{Z}$. By a special case of the Cartan–Dieudonné theorem (Gallier & Gallier, 2011), any 3D rotation can be expressed as a composition of two reflections across planes intersecting along the axis of rotation, where the rotation angle is twice the angle between the two planes. Consequently, we just need to search for the symmetry groups generated by at most three reflections. Each reflection is represented using the Hesse normal form (Bôcher, 1915; Duda & Hart, 1972), enabling efficient and parallelizable symmetry search (Je et al., 2024). When multiple symmetries exist, we select the one corresponding to the fundamental domain with the smallest cardinality. In practice, if a symmetry group is generated by fewer than three reflections, we pad the remaining slots with a special symbol to maintain consistent data dimensionality.

Since each part is generated separately, symmetry is enforced at the part level. During part generation, we sample only the fundamental domain $\mathbf{d}$ and recover the full part by applying the learned symmetry group as defined in Eq. (9). This approach reduces the number of points to generate and, more importantly, guarantees symmetry in the resulting shapes.

To learn the distribution of symmetry groups present in the dataset, we first construct a metric space $(\mathcal{M}, d_{\mathcal{M}})$ of reflectional symmetries following Je *et al.* (Je et al., 2024). We then obtain the ground truth for symmetry groups $\mathcal{S}$ using mean-shift clustering (Mitra et al., 2006; Fukunaga & Hostetler, 1975; Comaniciu & Meer, 2002), a nonparametric method based on gradient ascent over a density

function in $\mathcal{M}$ (see Appendix B.1 for more details). Finally, we leverage a diffusion model (Ho et al., 2020) to learn $p(\mathcal{S})$.

More specifically, we apply the diffusion process (Eq. (26) in Appendix B.3) to $\mathcal{S}$, and note that each transition kernel $q(\mathcal{S}_t \mid \mathcal{S}_{t-1}) \sim \mathcal{N}(\mu_t \mathcal{S}_{t-1}, \sigma_t I)$ is Gaussian. Therefore, intermediate samples $\mathcal{S}_t$ can be expressed in closed form as

$$\mathcal{S}_t = \left(\prod_{i=1}^{t} \mu_t\right) \mathcal{S}_0 + \sqrt{\sum_{i=1}^{t} \sigma_i^2 \prod_{j=i+1}^{t} \mu_j^2}\, \boldsymbol{\epsilon}, \tag{11}$$

where $\boldsymbol{\epsilon} \sim \mathcal{N}(\mathbf{0}, I)$. Setting $\mu_t := 1$ and $\gamma_t := \sqrt{\sum_{i=1}^{t} \sigma_i^2}$ simplifies this to $\mathcal{S}_t = \mathcal{S}_0 + \gamma_t \boldsymbol{\epsilon}$. Thus, the distribution of $\mathcal{S}_t$ becomes a convolution:

$$p(\mathcal{S}_t) = (p_0 * \varphi_{\mathbf{0},\gamma_t})(\mathcal{S}_t) = \int p_0(\mathcal{S})\, \varphi_{\mathbf{0},\gamma_t}(\mathcal{S}_t - \mathcal{S})\, \mathrm{d}\mathcal{S} = \int p_0(S)\, \varphi_{\mathcal{S},\gamma_t}(\mathcal{S}_t)\, \mathrm{d}\mathcal{S}, \tag{12}$$

where $p_0$ is the distribution of $\mathcal{S}_0$, $\varphi_{\mathcal{S},\gamma_t}$ is the probability density function for $\mathcal{N}(\mathcal{S}, \gamma_t I)$, and $*$ denotes convolution. With this, we train a diffusion model (Ho et al., 2020) $s_{\boldsymbol{\zeta}}(\mathcal{S}, t)$ parametrized by $\boldsymbol{\zeta}$ to approximate the time-dependent score function (Song et al., 2020b) (*i.e.* the gradient of the log-density of noisy data) via an empirical approximation (Je et al., 2024):

$$s_{\boldsymbol{\zeta}}(\mathcal{S}, t) \approx \nabla_{\mathcal{S}} \log p(\mathcal{S}_t) \approx \frac{1}{\gamma_t^2} \left( \frac{\sum_{R \in \mathcal{M}} \varphi_{R,\gamma_t}(\mathcal{S})\, R}{\sum_{R \in \mathcal{M}} \varphi_{R,\gamma_t}(\mathcal{S})} - \mathcal{S} \right). \tag{13}$$

Since symmetry can vary across shapes, in practice, we condition the diffusion model on the shape latent $\mathbf{z}$, modeling the symmetry group distribution as $p_{\boldsymbol{\zeta}}(\mathcal{S} \mid \mathbf{z})$.

After training, symmetries can be sampled from $s_{\boldsymbol{\zeta}}(\mathcal{S}, t)$ through annealed stochastic gradient Langevin dynamics (Song & Ermon, 2019): We initialize $\mathcal{S}_\tau^{(0)} \sim \mathcal{N}(\mathbf{0}, I)$ and sequentially sample from noise-perturbed distributions $s_{\boldsymbol{\zeta}}(\mathcal{S}, t)$ for $t = \tau, \tau - 1, \ldots, 1$ using $L$ Langevin steps:

$$\mathcal{S}_t^{(0)} \leftarrow \begin{cases} \boldsymbol{\epsilon}_\tau^{(0)} & \text{if } t = \tau \\ \mathcal{S}_{t+1}^{(L)} & \text{otherwise} \end{cases} ; \tag{14}$$

$$\mathcal{S}_t^{(i+1)} \leftarrow \mathcal{S}_t^{(i)} + \beta_t s_{\boldsymbol{\zeta}}\left(\mathcal{S}_t^{(i)}, t\right) + \sqrt{2\beta_t}\, \boldsymbol{\epsilon}_t^{(i)}, \quad i = 0, 1, \ldots, L, \tag{15}$$

where $\boldsymbol{\epsilon}_t^{(i)} \sim \mathcal{N}(\mathbf{0}, I)$, and $\beta_t$ is the step size. The final sample $\mathcal{S}_1^{(L)}$ is the generated symmetry.

Notably, substituting Eq. (13) into Eq. (15) and setting $\beta_t := \gamma_t^2$ yields the following update rule:

$$\mathcal{S}_t^{(i+1)} \leftarrow \mathcal{S}_t^{(i)} + \frac{\beta_t}{\gamma_t^2} \left( \frac{\sum_{T \in \mathcal{M}} \varphi_{T,\gamma_t}\left(\mathcal{S}_t^{(i)}\right) T}{\sum_{T \in \mathcal{M}} \varphi_{T,\gamma_t}\left(\mathcal{S}_t^{(i)}\right)} - \mathcal{S}_t^{(i)} \right) + \sqrt{2\beta_t}\, \boldsymbol{\epsilon} \stackrel{\beta_t := \gamma_t^2}{=} \frac{\sum_{T \in \mathcal{M}} \varphi_{T,\gamma_t}\left(\mathcal{S}_t^{(i)}\right) T}{\sum_{T \in \mathcal{M}} \varphi_{T,\gamma_t}\left(\mathcal{S}_t^{(i)}\right)} + \sqrt{2}\gamma_t \boldsymbol{\epsilon}. \tag{16}$$

This update rule resembles mean-shift clustering (Eq. (22) in Appendix B.1), except it assumes an infinite neighborhood $B(\mathcal{S}_t) = \mathcal{M}$ and uses a Gaussian kernel with bandwidth $\gamma_t$:

$$K_t(\mathcal{S}) := \frac{1}{\sqrt{2\pi}\gamma_t} e^{-\frac{\|\mathcal{S}\|^2}{2}}. \tag{17}$$

Another difference is the injected noise $\sqrt{2}\gamma_t \boldsymbol{\epsilon}$ in Eq. (16), which has been shown to improve robustness, sample quality, and mitigate mode collapse (*e.g.* Je et al., 2024).

Figure 3 shows examples of identified symmetry group generators. Most parts exhibit symmetry of a single reflection. In addition, we note that classical symmetries, such as reflectional and rotational, are special cases of our formulation, where the symmetry group $\mathcal{S}$ is generated by a single transformation. Our approach generalizes this by allowing symmetries composed of multiple transformations, such as sequential reflections (*e.g.*, chair legs in (e)) or a rotation followed by a reflection (*e.g.*, car wheels). While our model does not explicitly capture circular symmetry, it can be approximated using discrete rotations with a minimum angle of $\frac{\pi}{18}$ (*e.g.*, chair seat in (f)).

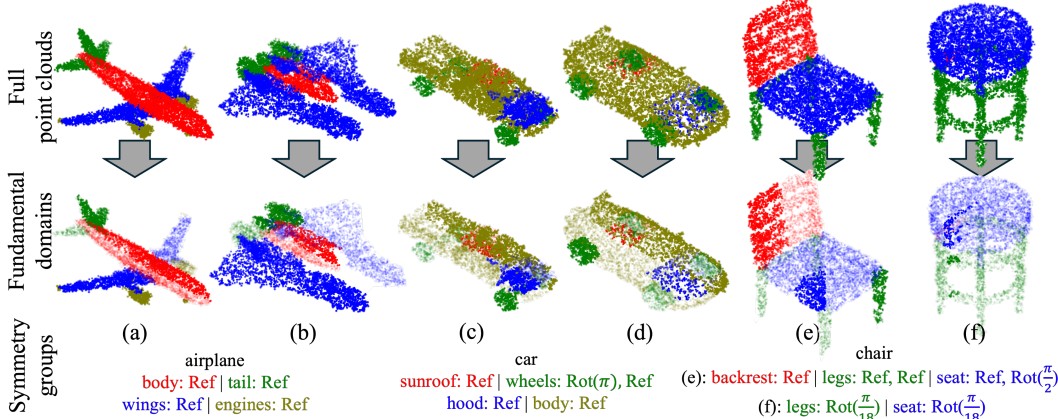

Figure 3: Point cloud airplanes, cars, and chairs with identified symmetry groups and corresponding fundamental domains for each color-coded part. Ref denotes reflection; Rot($\alpha$) denotes rotation by angle $\alpha$. Our symmetry formulation allows greater flexibility by supporting symmetries composed of multiple transformations, such as two reflections (chair legs (e)) or a rotation followed by a reflection (car wheels). Circular symmetry is approximated via small-angle rotations (chair seat (f)).

## 3.3 PART GENERATION

Given the identified symmetry groups $\mathcal{S}_j$ and corresponding fundamental domains $\mathbf{d}_j$ for each part $\mathbf{p}_j$, we model the part distribution via the distribution of fundamental domains. Since each full part $\mathbf{p}_j$ can be reconstructed by applying $\mathcal{S}_j$ to $\mathbf{d}_j$ (Eq. (9)), we have

$$p_{\boldsymbol{\xi}}\left(\mathbf{p}_j \mid \mathcal{S}_j, \mathbf{z}\right) = p_{\boldsymbol{\xi}}\left(\mathbf{d}_j \mid \mathcal{S}_j, \mathbf{z}\right). \tag{18}$$

To learn $p_{\boldsymbol{\xi}}\left(\mathbf{d}_j \mid \mathcal{S}_j, \mathbf{z}\right)$, we employ a transformer-based diffusion model (Mo et al., 2023; Peebles & Xie, 2023; Vaswani et al., 2017), conditioned on the shape latent $\mathbf{z}$ and symmetry group $\mathcal{S}_j$. The model operates directly on voxelized point clouds, using 3D positional and patch embeddings to capture spatial context. Unlike U-Net architectures (Ho et al., 2020), which face limitations in scalability and spatial coherence, our transformer backbone incorporates 3D window attention, reducing computational cost while preserving local structure. This design yields higher efficiency, scalability, and fidelity in point cloud generation (Mo et al., 2023).

## 3.4 PART ASSEMBLY

To assemble the full point cloud, we learn assemblers $T_j$ that correctly transform each part $\mathbf{p}_j$. We first train a part encoder $q_{\boldsymbol{\phi}}\left(\mathbf{w}_j \mid \mathbf{p}_j, \mathbf{z}\right)$ to obtain compact representations $\mathbf{w}_j$, then train a diffusion model $p_{\boldsymbol{\psi}}\left(T_j \mid \mathbf{w}_j, \mathbf{p}_j, \mathcal{S}_j, \mathbf{z}\right)$ to model the assembler distribution $p_{\boldsymbol{\psi}}\left(T_j \mid \mathbf{w}_j, \mathbf{p}_j, \mathcal{S}_j, \mathbf{z}\right)$. To simplify learning, we assume that $\mathbf{w}_j$ captures all necessary information about $\mathbf{p}_j$ and $\mathcal{S}_j$, making $T_j$ conditionally independent of them given $\mathbf{w}_j$. Consequently, the diffusion model is conditioned only on $\mathbf{w}_j$ and the global shape latent $\mathbf{z}$:

$$p_{\boldsymbol{\psi}}\left(T_j \mid \mathbf{w}_j, \mathbf{z}\right) \approx p_{\boldsymbol{\psi}}\left(T_j \mid \mathbf{w}_j, \mathbf{p}_j, \mathcal{S}_j, \mathbf{z}\right). \tag{19}$$

This assumption decouples geometric modeling from transformation learning, reducing complexity and allowing the model to focus on learning flexible, generalizable spatial configurations.

Our assembler diffusion model (Tang et al., 2024) employs a U-Net backbone (Ronneberger et al., 2015) augmented with skip connections and cross-attention layers (Vaswani et al., 2017). The cross-attention modules integrate contextual information from both part latents and the global shape latent, enabling the model to generate transformations that are coherent at both local and global levels. This architecture effectively handles diverse part configurations and generalizes well to novel shapes. The hierarchical structure of U-Net further supports multi-scale spatial reasoning, which is essential for accurate 3D part placement.

Table 1: Quantitative comparison of point cloud generation. PA denotes part awareness; SA denotes symmetry awareness. Our Quartet is the only model that supports both, achieving significant improvements over most baselines and setting a new state of the art.

| Model | PA | SA | Airplane | | | | Car | | | | Chair | | | |
|---|---|---|---|---|---|---|---|---|---|---|---|---|---|---|
| | | | 1-NNA ($\downarrow$) | | SDI ($\downarrow$) | | 1-NNA ($\downarrow$) | | SDI ($\downarrow$) | | 1-NNA ($\downarrow$) | | SDI ($\downarrow$) | |
| | | | CD | EMD | CD | EMD | CD | EMD | CD | EMD | CD | EMD | CD | EMD |
| Training set | | | 64.4 | 64.1 | 0.954 | 4.90 | 51.3 | 54.8 | 7.49 | 1.18 | 51.7 | 50.0 | 5.56 | 1.68 |
| PointFlow (Yang et al., 2019) | ✗ | ✗ | 75.7 | 70.7 | 3410 | 782 | 62.8 | 60.6 | 679 | 347 | 58.1 | 56.3 | 7290 | 530 |
| ShapeGF (Cai et al., 2020) | ✗ | ✗ | 81.2 | 80.9 | 332 | 98.6 | 58.0 | 61.3 | 645 | 40.9 | 61.8 | 57.2 | 1100 | 101 |
| DPF-Net (Klokov et al., 2020) | ✗ | ✗ | 75.2 | 65.6 | 4256 | 245 | 62.0 | 58.5 | 827 | 452 | 62.4 | 54.5 | 5234 | 245 |
| SetVAE (Kim et al., 2021) | ✗ | ✗ | 76.5 | 67.7 | 2830 | 824 | 58.8 | 60.6 | 1240 | 327 | 59.9 | 59.9 | 5320 | 673 |
| DPC (Luo & Hu, 2021) | ✗ | ✗ | 76.4 | 86.9 | 187 | 44.2 | 60.1 | 74.8 | 217 | 30.3 | 68.9 | 80.0 | 335 | 50.6 |
| PVD (Zhou et al., 2021a) | ✗ | ✗ | 73.8 | 64.8 | 150 | 42.0 | 56.3 | 53.3 | 213 | 31.2 | 54.6 | 53.8 | 275 | 58.4 |
| LION (Vahdat et al., 2022) | ✗ | ✗ | 67.4 | 61.2 | 97.2 | 40.6 | 53.7 | 52.3 | 168 | 30.8 | 53.4 | 51.1 | 201 | 55.2 |
| SPAGHETTI (Hertz et al., 2022) | ✓ | ✗ | 78.2 | 77.0 | 1530 | 529 | 72.3 | 71.0 | 581 | 284 | 70.7 | 69.0 | 5930 | 582 |
| DiT-3D (Mo et al., 2023) | ✗ | ✗ | 64.7 | 60.3 | 105 | 42.4 | 52.7 | **50.2** | 206 | 327 | 52.5 | 53.1 | 235 | 49.0 |
| SALAD (Koo et al., 2023) | ✓ | ✗ | 73.9 | 71.1 | 198 | 45.1 | 59.2 | 57.2 | 236 | 29.4 | 57.8 | 58.4 | 308 | 52.6 |
| FrePolad (Zhou et al., 2024) | ✗ | ✗ | 65.3 | 62.1 | 94.1 | 38.1 | 52.4 | 53.2 | 173 | 29.6 | 51.9 | **50.3** | 252 | 50.9 |
| Quartet (ours) | ✓ | ✓ | **63.3** | **59.7** | **25.7** | **1.87** | **50.1** | 51.8 | **25.7** | **2.28** | **51.6** | 53.7 | **28.9** | **2.86** |

For the part encoder, we adopt the SVAE architecture (see Sec. 3.1) with *equivariance fine-tuning* (*EFT*): During training, random rigid transformations, including translations, rotations, and reflections, are applied to each part, and the encoder is encouraged to produce the latent transformed accordingly. This promotes geometry-aware but pose-invariant part embeddings, enhancing their suitability for the part assembly task. As a result, the diffusion model receives more stable and semantically meaningful latents, leading to more accurate and coherent assembler predictions.

## 4 EXPERIMENT

### 4.1 DATASET

We evaluate our method on the ShapeNetPart dataset (Yi et al., 2016), a subset of ShapeNet (Chang et al., 2015) with semantic part annotations. We focus on three representative categories: airplanes (body, tail, wings, engines), cars (sunroof, wheels, hood, body), and chairs (backrest and armrests, legs, seat). Segmentation examples are shown in Fig. 3. We use the official train/validation/test split provided with the dataset.

Following common practice, we use point clouds with 2048 points. Each part is resized to match the mean number of points per part within its category. During preprocessing, parts with more points are randomly subsampled, while those with fewer are upsampled via random duplication. This normalization ensures consistent input dimensions across training batches, enhancing the stability and efficiency of both the encoder and generative models.

### 4.2 EVALUATION METRICS

Following prior work (*e.g.* Zhou et al., 2024), we assess the quality and diversity of generated point clouds using 1-nearest neighbor (1-NNA) (Lopez-Paz & Oquab, 2016), a retrieval-based metric computed with Chamfer Distance (CD) and Earth Mover's Distance (EMD). 1-NNA measures how often a shape's nearest neighbor belongs to the same distribution (Yang et al., 2019). A balanced score near 50% indicates close alignment between generated and real shape distributions.

To assess whether a generated 3D shape is symmetric, we introduce the *symmetry discrepancy index (SDI)*, which quantifies how well a 3D shape $\mathbf{p}$ conforms to a given symmetry group $\mathcal{S}$: For a normalized shape $\mathbf{p}$, SDI is defined as the distance $d$ — either CD or EMD — between $\mathbf{p}$ and the shape reconstructed from its fundamental domain $\mathbf{d}$ under $\mathcal{S}$:

$$\mathcal{L}_{\text{SDI}}(\mathbf{p}) \coloneqq d\left(\mathbf{p}, \mathcal{S}\mathbf{d}\right). \tag{20}$$

Lower SDI values indicate stronger symmetry. For methods without explicit symmetry modeling, we compute SDI using the simplest yet most common symmetry: reflection across the vertical bisector plane. For readability, we report SDI-CD scaled by 10 and SDI-EMD scaled by $10^3$.

Table 2: Per-part SDI-CD (↓) for point cloud generation. With explicit symmetry enforcement, the Quartet achieves significantly lower SDI scores, indicating stronger symmetry in generated parts.

| Model | Airplane | | | | Car | | | | Chair | | |
|---|---|---|---|---|---|---|---|---|---|---|---|
| | body | tail | wings | engines | roof | wheels | hood | body | back | legs | seat |
| Training set | 0.877 | 0.700 | 0.883 | 0.717 | 5.25 | 3.06 | 3.15 | 5.27 | 4.11 | 2.65 | 3.04 |
| SPAGHETTI (Hertz et al., 2022) | 318 | 268 | 417 | 125 | 258 | 339 | 127 | 417 | 2857 | 2948 | 1358 |
| SALAD (Koo et al., 2023) | 39.1 | 26.2 | 30.5 | 18.5 | 98.2 | 68.5 | 152 | 256 | 163 | 93.5 | 104 |
| Quartet (ours) | **7.86** | **4.08** | **7.76** | **5.84** | **9.72** | **4.10** | **5.76** | **9.90** | **10.3** | **6.19** | **10.4** |

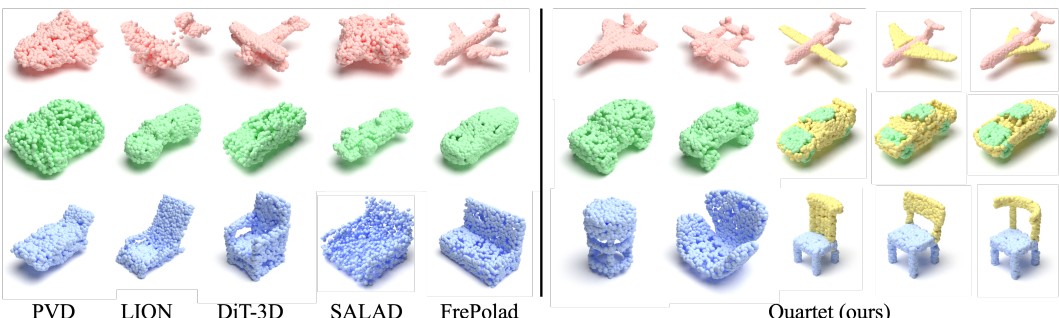

PVD    LION    DiT-3D    SALAD    FrePolad    Quartet (ours)

Figure 4: Point cloud generation. Samples from the Quartet are visually appealing, diverse, and exhibit strong structural consistency. The last three columns illustrate targeted manipulation.

## 4.3 POINT CLOUD GENERATION

We benchmark the Quartet against several competitive 3D generative models. Implementation and training details are provided in Appendix C, and additional experimental results can be found in Appendix D. Quantitative results based on 1-NNA and SDI are reported in Tab. 1. Notably, the Quartet is the only method that explicitly models both part structure and symmetry. Per-part SDI scores for part-aware models are presented in Tab. 2. The Quartet consistently outperforms state-of-the-art methods, achieving high fidelity and diversity while closely matching the real shape distribution. Remarkably, due to explicit symmetry enforcement, the Quartet achieves significantly lower SDI scores, indicating strong alignment with ideal symmetric structures. Qualitative results in Fig. 4 show that the Quartet generates visually coherent and diverse point clouds obeying symmetry across all three object categories, demonstrating its effectiveness.

**Targeted manipulation**    The Quartet models the distribution of each part separately, enabling targeted part-level manipulation while guaranteeing disentanglement. In Fig. 1 and in the last three columns of Fig. 4, we present point cloud samples in which the yellow-highlighted parts vary while the remaining structure is held fixed.

## 4.4 ABLATION STUDY

Our Quartet consists of four diffusion models responsible for shape latents, symmetries, parts, and assemblers. As an ablation study, in Tab. 3, we evaluate several simplified variants with some diffusions or key techniques removed:

- NoSVAE: replaces SVAEs (Sec. 3.1) with standard VAEs for full point cloud and part encoding.
- NoEFT: removes the equivariance fine-tuning (Sec. 3.4) applied to the part SVAE.
- Trio Var. 1: replaces shape latent diffusion with a simple Gaussian prior;
- Trio Var. 2: removes the symmetry diffusion with part diffusion generating the full part directly;
- Duet Var. 1: removes both shape latent and symmetry diffusions;
- Duet Var. 2: removes symmetry diffusion and merges part and assembler diffusions into a single full-shape diffusion;
- Solo: uses a single diffusion generating full point clouds.

Table 3: Ablation study. L denotes latent diffusion, S symmetry diffusion, and P part and assembler diffusions. All four members in the Quartet are essential; removing any degrades performance.

| Variation | L | S | P | Airplane | | | | Car | | | | Chair | | | |
|---|---|---|---|---|---|---|---|---|---|---|---|---|---|---|---|
| | | | | 1-NNA ($\downarrow$) | | SDI ($\downarrow$) | | 1-NNA ($\downarrow$) | | SDI ($\downarrow$) | | 1-NNA ($\downarrow$) | | SDI ($\downarrow$) | |
| | | | | CD | EMD | CD | EMD | CD | EMD | CD | EMD | CD | EMD | CD | EMD |
| Solo | ✗ | ✗ | ✗ | 69.2 | 64.2 | 154 | 44.2 | 55.5 | 53.6 | 241 | 351 | 53.2 | 53.8 | 295 | 52.8 |
| Duet Var. 1 | ✗ | ✗ | ✓ | 76.3 | 82.1 | 927 | 614 | 70.2 | 72.9 | 522 | 460 | 68.3 | 67.2 | 3261 | 615 |
| Duet Var. 2 | ✓ | ✗ | ✗ | 66.8 | 63.3 | 103 | 42.4 | 52.3 | 52.7 | 192 | 28.1 | 53.2 | 51.6 | 184 | 49.2 |
| Trio Var. 1 | ✗ | ✓ | ✓ | 85.1 | 85.8 | 3516 | 1450 | 83.9 | 83.6 | 836 | 326 | 92.5 | 85.2 | 6286 | 562 |
| Trio Var. 2 | ✓ | ✗ | ✓ | **63.1** | 63.6 | 95 | 39.2 | **49.8** | 52.4 | 205 | 32.3 | 52.3 | 53.9 | 174 | 52.5 |
| NoSVAE | ✓ | ✓ | ✓ | 64.3 | 61.8 | **25.2** | 2.51 | 51.6 | 52.0 | 25.9 | 2.55 | 52.6 | 53.9 | 29.3 | 3.15 |
| NoEFT | ✓ | ✓ | ✓ | 63.9 | 62.4 | 32.7 | 9.27 | 51.8 | 52.0 | 27.2 | 5.98 | 52.5 | 54.0 | 30.2 | 4.15 |
| Quartet | ✓ | ✓ | ✓ | 63.3 | **59.7** | 25.7 | **1.87** | 50.1 | **51.8** | 25.7 | **2.28** | **51.6** | 53.7 | **28.9** | **2.86** |

Our results show that all four diffusion models are indispensable members of the Quartet, each contributing to the generation of high-quality, diverse point clouds with strong symmetry. Two key observations emerge: First, comparing Solo and Duet Var. 1, we see that generating parts without a central shape latent significantly degrades performance. This echoes previous findings that a simple Gaussian prior is insufficient for capturing the structural complexity of 3D shapes (Vahdat et al., 2021; Tomczak & Welling, 2018; Rosca et al., 2018; Zhou et al., 2024; Vahdat et al., 2022). The performance drops further in Trio Var. 1, where symmetry diffusion is added, likely compounding the mismatch between prior and part representations. Second, comparing Trio Var. 2 and the full Quartet, we observe that while symmetry enforcement substantially improves SDI, it does not consistently improve 1-NNA – even with latent diffusion present. This may be due to the added constraint or increased learning complexity introduced by enforcing symmetry.

## 4.5 RUNTIME ANALYSIS

Training the Quartet on each object category takes approximately 50 hours on a single GPU. Figure 1 presents the generation performance *vs.* training time for models trained on the airplane category. Some baselines require longer training due to joint optimization (Yang et al., 2019), operation on full point clouds (Zhou et al., 2021a; Luo & Hu, 2021), or the use of complex, high-dimensional latent spaces (Vahdat et al., 2022). In contrast, despite using four diffusions, the Quartet trains faster than most baselines. This is because the shape latent, symmetry, and assembler diffusions operate on low-dimensional representations ($124$-, $12M$-, and $9M$-dimensional, respectively, where $M$ is the number of parts). The majority of training time is spent on part diffusion, which is accelerated using an efficient transformer-based diffusion (Mo et al., 2023) applied to fundamental domains that are typically only half or a quarter the size of the full parts.

## 5 CONCLUSION AND DISCUSSION

We presented the Quartet of Diffusions, a structure-aware framework for 3D point cloud generation that explicitly models part composition and symmetry. At its current stage, our work focuses on unconditional generation, but it highlights the importance of integrating symmetry- and part-based reasoning into structure-aware models. The model's ability to disentangle and manipulate individual parts makes it well-suited for interactive shape editing and user-guided design applications.

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

# Quartet of Diffusions: Structure-Aware Point Cloud Generation through Part and Symmetry Guidance

## Appendix

## A  RELATED WORK ON SYMMETRY DETECTION

Symmetry is a fundamental geometric property commonly observed in both natural and human-made objects. As a result, detecting and leveraging symmetry has long been a central problem in the fields of computer vision and computer graphics (Mitra et al., 2013a). Early work in this area focused on detecting exact symmetries in 2D or 3D planar point sets (Atallah, 1985; Wolter et al., 1985). However, the reliance on exact symmetry limits the practicality of these methods, as real-world objects often exhibit only approximate symmetry due to noise, occlusion, or design variations.

Traditional voting-based approaches, such as those based on the Hough transform (Illingworth & Kittler, 1988), attempt to accumulate votes for potential symmetries from point correspondences. While effective in idealized settings, these methods are known to be sensitive to noise and often produce unreliable results when applied to imperfect or incomplete data.

To address these limitations, Mitra *et al.* (Mitra et al., 2006) proposed a more robust technique that replaces voting with mean-shift clustering (Fukunaga & Hostetler, 1975; Comaniciu & Meer, 2002) in a transformation space (for more details see Appendix B.1). This method searches for modes in the space of rigid transformations, making it more resilient to noise and better suited for partial symmetries. Their approach laid the groundwork for several subsequent improvements (Pauly et al., 2008; Shi et al., 2016), which further enhanced the robustness and generality of symmetry detection.

In recent years, learning-based methods have also been introduced (Ji & Liu, 2019; Zhou et al., 2021b; Gao et al., 2020), leveraging neural networks to detect symmetry directly from 3D data. These methods benefit from data-driven representations and have shown improved generalization across object categories and varying conditions.

Building on these trends, Je *et al.* (Je et al., 2024) proposed a hybrid approach that combines elements of both traditional and learning-based techniques. Their method redefines the symmetry space and applies Langevin dynamics—a sampling technique from generative modeling—to iteratively refine symmetry estimates. This formulation provides both robustness and efficiency and serves as a strong foundation for the symmetry detection component in our framework.

## B  BACKGROUND

### B.1  SYMMETRY DETECTION BY MEAN-SHIFT CLUSTERING

A common approach to detecting symmetry in 3D objects involves using the Hough transform (Illingworth & Kittler, 1988) to vote on the parameters of potential symmetry planes (Mitra et al., 2013a; 2006). In this framework, each pair of points in the shape casts a vote for a candidate symmetry in a predefined transformation space $\mathcal{M}$. For example, in the case of reflectional symmetry, a point pair $\mathbf{a}$ and $\mathbf{b}$ casts a vote for the plane that passes through their midpoint $\frac{\mathbf{a}+\mathbf{b}}{2}$ and has a normal vector given by $\frac{\mathbf{a}-\mathbf{b}}{\|\mathbf{a}-\mathbf{b}\|}$ – the direction that would reflect one point onto the other.

In an ideal setting, where the object and its symmetries are exact, votes from all symmetric point pairs would concentrate on discrete points in $\mathcal{M}$. In such cases, detecting symmetry reduces to identifying the peak with the highest vote count. However, most real-world objects and 3D datasets only exhibit approximate symmetry. As a result, the votes form a smooth, continuous distribution in $\mathcal{M}$ rather than sharp peaks. Detecting symmetry in this context thus requires identifying clusters in $\mathcal{M}$ that correspond to the most prominent approximate symmetries.

Mitra *et al.* (Mitra et al., 2006) proposes to use the mean-shift clustering (Fukunaga & Hostetler, 1975; Comaniciu & Meer, 2002), a nonparametric clustering method based on gradient ascent on a

density function $p(S)$ in $\mathcal{M}$ defined as

$$p(S) := \sum_{R \in \mathcal{M}} K\left(\frac{S - R}{h}\right), \tag{21}$$

where $K$ is a kernel function (*e.g.*, Gaussian or Epanechnikov kernel (Epanechnikov, 1969)) with bandwidth $h$. The significant modes of $p$, and hence the significant symmetries, can be determined using gradient ascent: The algorithm first initializes from any of the candidate transformation $S_0 \in \mathcal{M}$ and performs the following iteration until convergence:

$$S_{t+1} \leftarrow \frac{\sum_{R \in B(S_t)} K\left(\frac{S_t - R}{h}\right) R}{\sum_{R \in B(S_t)} K\left(\frac{S_t - R}{h}\right)}, \tag{22}$$

where $B(S_t)$ is a neighborhood of $S_t$.

While it has shown promising results in detecting symmetries (*e.g.*, Chang et al., 2015), it may fail under noisy shapes or noisy transformation space (Je et al., 2024). Inspired by Je *et al.* (Je et al., 2024), we leverage a diffusion process (Ho et al., 2020) to establish the connection between the iteration in Eq. (22) and the stochastic gradient Langevin dynamics (Welling & Teh, 2011), where stochastic noise is injected to improve the sample quality and robustness.

## B.2 Variational Autoencoder

We use *variational autoencoders (VAEs)* (Kingma & Welling, 2013) as our latent distribution model as it provides access to a low-dimensional latent space and has been successfully applied to generate point clouds (Li et al., 2022a; Wang et al., 2020; Li et al., 2022b; Zhou et al., 2024). VAEs are probabilistic generative models that can model a probability distribution of a given dataset $\mathcal{X}$.

Starting with a known prior distribution $p(\mathbf{z})$ of shape latents $\mathbf{z} \in \mathbb{R}^{\mathbf{z}}$, the parametric decoder of a VAE models the conditional distribution $p_{\boldsymbol{\eta}}(\mathbf{x} \mid \mathbf{z})$ parametrized by $\boldsymbol{\eta}$. However, training the decoder to maximize the likelihood of data is not possible as

$$p(\mathbf{x}) = \int p_{\boldsymbol{\eta}}(\mathbf{x} \mid \mathbf{z}) \, p(\mathbf{z}) \, \mathrm{d}\mathbf{z} \tag{23}$$

is intractable. Instead, a parametric encoder $p_{\boldsymbol{\eta}}(\mathbf{x} \mid \mathbf{z})$ is used to approximates the posterior distribution. Both networks are jointly trained to maximize a lower bound on the likelihood called the *evidence lower bound (ELBO)*:

$$\mathcal{L}_{\text{ELBO}}(\boldsymbol{\eta}; \mathbf{x}) := \mathbb{E}_{q_{\boldsymbol{\eta}}(\mathbf{z}|\mathbf{x})}\left[\log p_{\boldsymbol{\eta}}(\mathbf{x} \mid \mathbf{z})\right] - \mathcal{D}_{\text{KL}}\left(q_{\boldsymbol{\eta}}(\mathbf{z} \mid \mathbf{x}), p(\mathbf{z})\right), \tag{24}$$

where $\mathcal{D}_{\text{KL}}$ is the Kullback-Leibler divergence between the two distributions (Csiszár, 1975).

## B.3 Denoising Diffusion Probabilistic Model

Our generative framework leverages four *denoising diffusion probabilistic models (DDPMs or diffusion models)* (Sohl-Dickstein et al., 2015; Ho et al., 2020) to model distinct data distributions. Given a data sample $\mathbf{z} \sim p(\mathbf{z})$, diffusion models progressively corrupt $\mathbf{z} = \mathbf{z}_0$ into a noisy sample $\mathbf{z}_\tau$ through a Markovian *forward diffusion process*. At each time step $t = 1, 2, \ldots, \tau$, Gaussian noise is added according to a predefined variance schedule $\{\sigma_t\}_t$:

$$q\left(\mathbf{z}_{1:\tau} \mid \mathbf{z}_0\right) := \prod_{t=1}^{\tau} q(\mathbf{z}_t | \mathbf{z}_{t-1}) \tag{25}$$

$$q(\mathbf{z}_t \mid \mathbf{z}_{t-1}) \sim \mathcal{N}\left(\mu_t \mathbf{z}_{t-1}, \sigma_t I\right), \tag{26}$$

where $\mathcal{N}(\boldsymbol{\mu}, \boldsymbol{\sigma})$ denotes multivariate Gaussian distribution with mean $\boldsymbol{\mu}$ and variance $\boldsymbol{\sigma}$. In practice, we set $\mu_t := \sqrt{1 - \sigma_t}$. If $\tau$ is sufficiently large (*e.g.*, 1000 steps), $p(\mathbf{z}_\tau)$ will approach the standard Gaussian distribution $\mathcal{N}(0, I)$.

Diffusion models learn a *reverse process* $p_{\boldsymbol{\theta}}(\mathbf{z}_{t-1} \mid \mathbf{z}_t)$, parameterized by $\boldsymbol{\theta}$, which defines a Markovian denoising chain that inverts the forward diffusion. This process gradually transforms a sample

of standard Gaussian noise $\mathbf{z}_\tau$ back into a data sample $\mathbf{z}_0$:

$$p_{\boldsymbol{\theta}}(\mathbf{z}_{0:\tau}) := p(\mathbf{z}_\tau) \prod_{t=1}^{\tau} p_{\boldsymbol{\theta}}(\mathbf{z}_{t-1} \mid \mathbf{z}_t); \tag{27}$$

$$p_{\boldsymbol{\theta}}(\mathbf{z}_{t-1} \mid \mathbf{z}_t) \sim \mathcal{N}\left(\boldsymbol{\mu}_{\boldsymbol{\theta}}(\mathbf{z}_t, t), \varsigma_t^2 I\right), \tag{28}$$

where $\boldsymbol{\mu}_{\boldsymbol{\theta}}(\mathbf{z}_t, t)$ represents the predicted mean for the Gaussian distribution at time step $t$ and $\{\varsigma_t\}_t$ is another variance schedule.

DDPMs are trained by maximizing the variational lower bound of log-likelihood of the data $\mathbf{z}_0$ under $q(\mathbf{z}_0)$:

$$\mathbb{E}_{q(\mathbf{z}_0)}\left[\log p_{\boldsymbol{\theta}}(\mathbf{z}_0)\right] \geq \mathbb{E}_{q(\mathbf{z}_{0:\tau})}\left[\log \frac{p_{\boldsymbol{\theta}}(\mathbf{z}_{0:\tau})}{q(\mathbf{z}_{1:\tau} \mid \mathbf{z}_0)}\right]. \tag{29}$$

Expanding Eq. (29) with Eq. (27) and noticing that $p(\mathbf{z}_\tau)$ and $q(\mathbf{z}_{1:\tau} \mid \mathbf{z}_0)$ are constant with respect to $\boldsymbol{\theta}$, we obtain our objective function to maximize:

$$\mathbb{E}_{q(\mathbf{z}_0),q(\mathbf{z}_{1:\tau}|\mathbf{z}_0)}\left[\sum_{t=1}^{\tau} \log p_{\boldsymbol{\theta}}(\mathbf{z}_{t-1} \mid \mathbf{z}_t)\right]. \tag{30}$$

Since we can factor the joint posterior

$$q(\mathbf{z}_{1:\tau} \mid \mathbf{z}_0) = \prod_{t=1}^{\tau} q(\mathbf{z}_{t-1} \mid \mathbf{z}_t, \mathbf{z}_0) \tag{31}$$

and both $q(\mathbf{z}_{t-1} \mid \mathbf{z}_t, \mathbf{z}_0)$ and $p_{\boldsymbol{\theta}}(\mathbf{z}_{t-1} \mid \mathbf{z}_t)$ are Gaussian, maximizing Eq. (30) is equivalent as minimizing the following score matching objective for a parametric model $\boldsymbol{\epsilon}_{\boldsymbol{\theta}}(\mathbf{z}_t, t)$:

$$\mathcal{L}_{\text{diffusion}}(\boldsymbol{\theta}) := \mathbb{E}_{p(\mathbf{z}_0),t\sim\mathcal{U}(1,\tau),\boldsymbol{\epsilon}\sim\mathcal{N}(0,I)}\left[\|\boldsymbol{\epsilon} - \boldsymbol{\epsilon}_{\boldsymbol{\theta}}(\mathbf{z}_t, t)\|^2\right], \tag{32}$$

where $\mathcal{U}(1,\tau)$ is the uniform distribution on $\{1, 2, \ldots, \tau\}$. Intuitively, minimizing this loss corresponds to learning to predict the noise $\boldsymbol{\epsilon}$ required to denoise the diffused sample $\mathbf{z}_t$. Notably, the training objective of diffusion models closely aligns with estimating the gradient of the log data density – *i.e.*, the score function – as used in score-based energy models (Song et al., 2020a; Swersky et al., 2011; Hyvärinen et al., 2009; LeCun et al., 2006).

During inference, the network allows sampling through an iterative procedure since the learned distribution can be factorized as

$$p_{\boldsymbol{\theta}}\left(\{\mathbf{z}_t\}_{t=0}^{\tau}\right) = p\left(\mathbf{z}_\tau\right) p_{\boldsymbol{\theta}}\left(\mathbf{z}_{t-1} \mid \mathbf{z}_t\right) = p\left(\mathbf{z}_\tau\right) \prod_{t=1}^{\tau} p_{\boldsymbol{\theta}}\left(\mathbf{z}_{t-1} \mid \mathbf{z}_t\right) \tag{33}$$

for $p(\mathbf{z}_\tau) := \mathcal{N}(0, I)$.

## C  IMPLEMENTATION AND TRAINING DETAILS

Tables 4 and 5 provide the architecture and training details for the various VAE and diffusion models used in the Quartet. The hyperparameter $\lambda$ in Eq. (8), which controls the activation sparsity in the SVAE, is set to 0.005. The hyperparameter for the equivariance fine-tuning in the part VAE, discussed in Sec. 3.4, is set to 0.01. For all models, we employ a learning rate scheduler with a reduce-on-plateau policy, which decreases the learning rate by a factor of 10 if the loss does not improve for 10 consecutive epochs. All training runs converged successfully by the end of training.

## D  MORE EXPERIMENTAL RESULTS

Table 6 and Figs. 5 and 6 present more results for the point cloud generation experiment in Sec. 4.3.

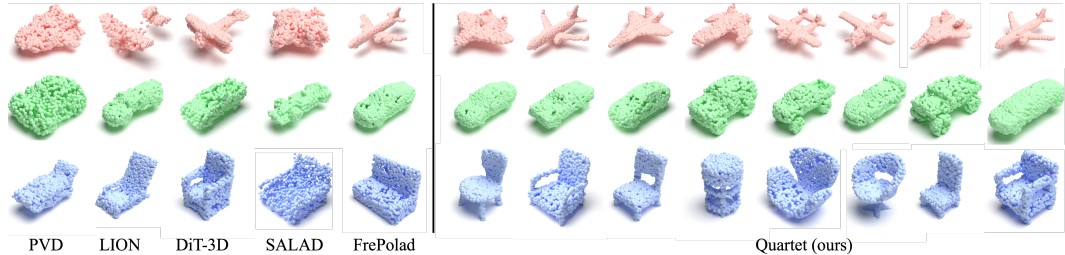

PVD  LION  DiT-3D  SALAD  FrePolad                    Quartet (ours)

Figure 5: Full point cloud generation results. Samples from the Quartet are visually appealing, diverse, and exhibit strong structural and symmetry consistency.

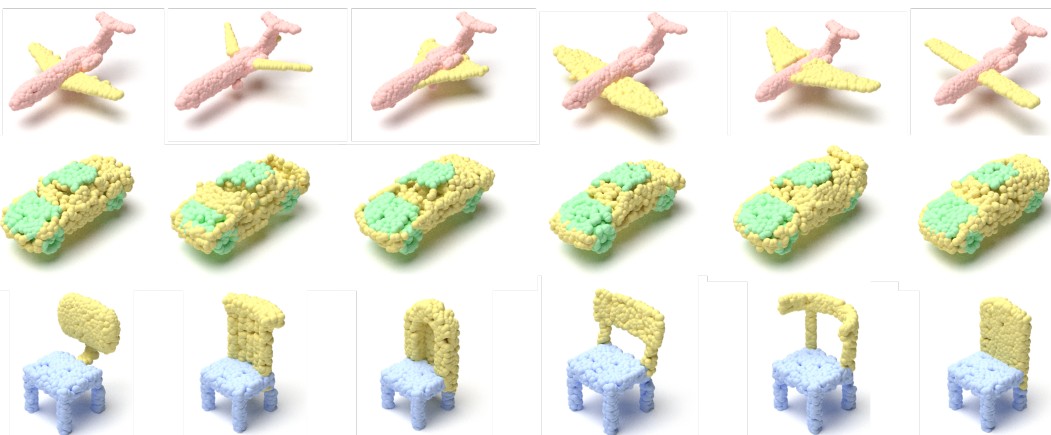

Figure 6: More targeted manipulation results. We vary the yellow-highlighted parts while holding the remaining structure fixed.

Table 4: Model architecture details for the Quartet. $M$ denotes the number of parts for each object category ($M = 4$ for airplanes and cars; $M = 3$ for chairs).

| Model | Backbone | Data dimensionality | Latent dimensionality |
|---|---|---|---|
| Point cloud SVAE | PVCNN (Liu et al., 2019) | $2048 \times 3$ | 128 |
| Shape latent diffusion | U-Net (Ronneberger et al., 2015) | 128 | - |
| Symmetry diffusion | ResNet (He et al., 2016) | $12 \times M$ | - |
| Part diffusion | Transformer (Vaswani et al., 2017) | $2048 \times 3$ | - |
| Part VAE | PVCNN (Liu et al., 2019) | $57 \times 3$ to $1024 \times 3$ | 128 |
| Assembler diffusion | U-Net (Ronneberger et al., 2015) | $9 \times M$ | - |

Table 5: Training details for the Quartet.

| Model | Batch size | Number of epoch | Learning rate |
|---|---|---|---|
| Point cloud SVAE | 64 | 1000 | $10^{-3}$ |
| Shape latent diffusion | 32 | 2000 | $10^{-4}$ |
| Symmetry diffusion | 32 | 2000 | $10^{-4}$ |
| Part diffusion | 16 | 2000 | $10^{-3}$ |
| Part VAE | 64 | 1000 | $10^{-5}$ |
| Assembler diffusion | 32 | 2000 | $10^{-4}$ |

Table 6: Quantitative comparison of point cloud generation. PA denotes part awareness; SA denotes symmetry awareness. Our Quartet is the only model that supports both, achieving significant improvements over most baselines and setting a new state of the art.

| Model | PA | SA | Airplane 1-NNA (↓) CD | EMD | Airplane SDI (↓) CD | EMD | Car 1-NNA (↓) CD | EMD | Car SDI (↓) CD | EMD | Chair 1-NNA (↓) CD | EMD | Chair SDI (↓) CD | EMD |
|---|---|---|---|---|---|---|---|---|---|---|---|---|---|---|
| Training set | | | 64.4 | 64.1 | 0.954 | 4.90 | 51.3 | 54.8 | 7.49 | 1.18 | 51.7 | 50.0 | 5.56 | 1.68 |
| r-GAN (Achlioptas et al., 2018) | ✗ | ✗ | 98.4 | 96.8 | 4519 | 1410 | 83.7 | 99.7 | 1053 | 362 | 94.5 | 99.0 | 7249 | 619 |
| l-GAN/CD (Achlioptas et al., 2018) | ✗ | ✗ | 87.3 | 94.0 | 3629 | 1353 | 68.6 | 83.8 | 918 | 352 | 66.5 | 88.8 | 7972 | 582 |
| l-GAN/EMD (Achlioptas et al., 2018) | ✗ | ✗ | 89.5 | 76.9 | 4129 | 914 | 71.9 | 64.7 | 982 | 313 | 71.2 | 66.2 | 7184 | 521 |
| PointFlow (Yang et al., 2019) | ✗ | ✗ | 75.7 | 70.7 | 3410 | 782 | 62.8 | 60.6 | 679 | 347 | 58.1 | 56.3 | 7290 | 530 |
| SoftFlow (Kim et al., 2020) | ✗ | ✗ | 76.1 | 65.8 | 3284 | 529 | 59.2 | 60.1 | 1549 | 428 | 64.8 | 60.1 | 5628 | 420 |
| ShapeGF (Cai et al., 2020) | ✗ | ✗ | 81.2 | 80.9 | 332 | 98.6 | 58.0 | 61.3 | 645 | 40.9 | 61.8 | 57.2 | 1100 | 101 |
| DPF-Net (Klokov et al., 2020) | ✗ | ✗ | 75.2 | 65.6 | 4256 | 245 | 62.0 | 58.5 | 827 | 452 | 62.4 | 54.5 | 5234 | 245 |
| SetVAE (Kim et al., 2021) | ✗ | ✗ | 76.5 | 67.7 | 2830 | 824 | 58.8 | 60.6 | 1240 | 327 | 59.9 | 59.9 | 5320 | 673 |
| DPC (Luo & Hu, 2021) | ✗ | ✗ | 76.4 | 86.9 | 187 | 44.2 | 60.1 | 74.8 | 217 | 30.3 | 68.9 | 80.0 | 335 | 50.6 |
| PVD (Zhou et al., 2021a) | ✗ | ✗ | 73.8 | 64.8 | 150 | 42.0 | 56.3 | 53.3 | 213 | 31.2 | 54.6 | 53.8 | 275 | 58.4 |
| LION (Vahdat et al., 2022) | ✗ | ✗ | 67.4 | 61.2 | 97.2 | 40.6 | 53.7 | 52.3 | 168 | 30.8 | 53.4 | 51.1 | 201 | 55.2 |
| SPAGHETTI (Hertz et al., 2022) | ✓ | ✗ | 78.2 | 77.0 | 1530 | 529 | 72.3 | 71.0 | 581 | 284 | 70.7 | 69.0 | 5930 | 582 |
| DiT-3D (Mo et al., 2023) | ✗ | ✗ | 64.7 | 60.3 | 105 | 42.4 | 52.7 | **50.2** | 206 | 327 | 52.5 | 53.1 | 235 | 49.0 |
| SALAD (Koo et al., 2023) | ✓ | ✗ | 73.9 | 71.1 | 198 | 45.1 | 59.2 | 57.2 | 236 | 29.4 | 57.8 | 58.4 | 308 | 52.6 |
| FrePolad (Zhou et al., 2024) | ✗ | ✗ | 65.3 | 62.1 | 94.1 | 38.1 | 52.4 | 53.2 | 173 | 29.6 | 51.9 | **50.3** | 252 | 50.9 |
| Quartet (ours) | ✓ | ✓ | **63.3** | **59.7** | **25.7** | **1.87** | **50.1** | 51.8 | **25.7** | **2.28** | 51.6 | 53.7 | **28.9** | **2.86** |

