# OpenReview forum: "Quartet of Diffusions: Structure-Aware Point Cloud Generation through Part and Symmetry Guidance"
_ICLR.cc/2026/Conference — ICLR 2026 Poster_

### Official Review · Reviewer_E7K3 · 2025-10-29

**Soundness:** 3
**Presentation:** 3
**Contribution:** 2
**Rating:** 6
**Confidence:** 4

**Summary:**

The paper proposes a part-level point cloud generation framework called the "Quartet of Diffusions," which integrates part and symmetry awareness into the generative process. The method uses four diffusion models to generate structure-aware 3D shapes by modeling shape latents, symmetries, parts, and assemblers. The Quartet framework is shown to outperform state-of-the-art methods in terms of diversity, quality, and symmetry.

**Strengths:**

- Part-level 3D generation is a critical research topic recently.
- Using separate diffusion models to represent parts and transformations between them is novel and interesting, ensuring global shape symmetry.
- The disentangled representation of parts, symmetries, and global structure offers fine-grained control over the generation process, allowing for targeted manipulation of individual parts.
- The method pipeline is complex but clearly written.

**Weaknesses:**

- Currently, part generation methods based on implicit representations (e.g., PartCrafter, PartPacker, OmniPart) have achieved very impressive results, almost reaching production-ready levels. However, this paper still insists on using point clouds as the 3D representation, and the experiments are only validated on the ShapeNet dataset. As this approach can only generate a limited number of points and coarse geometric details, this somewhat diminishes the contribution and novelty of the paper. Therefore, I recommend that the authors discuss in detail why they chose this approach and clarify the advantages of using point clouds as a 3D representation for structured 3D generation.
- The Quartet framework involves four separate diffusion models, which could introduce significant computational overhead during both training and inference, making it potentially less practical for real-time applications.
- Integration of four diffusion models will lead to accumulated errors, which are not analyzed.

**Questions:**

See the weaknesses

---

> ### Author Response · Authors · 2025-11-20
>
> Thank you for recognizing the strengths and providing insightful and valuable comments on our work. Below we address each point in the Weaknesses section.
>
> - **Choice of point clouds**: We thank the reviewer for this thoughtful comment. **Point clouds remain a standard and widely adopted 3D representation** in generative modeling, reconstruction, and robotics. Many recent frameworks, including autoencoder-, flow-, and diffusion-based approaches, operate in the point domain due to its **simplicity, differentiability, and flexibility** across shape categories. Moreover, surface reconstruction algorithms (e.g., Hanocka et al., 2020; Amin Mansour et al., 2024) can directly convert our generated point clouds into watertight meshes with high geometric fidelity. We will expand the discussion in the revised paper to highlight these practical advantages.
> Regarding **evaluation on ShapeNet**, we followed the **standard benchmark protocol** used by prior 3D point cloud generation baselines in Tab. 1. ShapeNet provides diverse part structures and well-annotated symmetry patterns, enabling direct and fair comparison with existing baselines. While extending experiments to more complex datasets is feasible, it would limit comparability since most prior works are not trained on them. That said, we agree that broader evaluation would provide additional insights, and we are happy to include results on additional categories or datasets in the revised version.
> - **Runtime efficiency**: We thank the reviewer for raising this important point. As reported in the runtime analysis in Sec. 4.5 and to the right of Fig. 1, our framework achieves **state-of-the-art results with competitive runtime performance**. Although the Quartet framework employs four diffusion components, three of them (shape latent, symmetry, and assembly) operate on *low-dimensional latent spaces*, resulting in minimal computational cost. The majority of runtime is attributed to the part diffusion, which we accelerate using a *lightweight transformer-based diffusion* applied to fundamental domains that are typically one-half or one-quarter the size of full parts. This design yields both efficiency and high fidelity, making the method more practical for real-time applications.
> - **Error analysis**: Thank you for the suggestion. While cascading multiple diffusion components could, in principle, introduce error propagation, our design minimizes this risk in two ways. First, the four diffusion processes operate on distinct, complementary representations: global shape latent, symmetry, parts, and assembly, rather than sequentially refining the same output. Each diffusion is trained independently with its own denoising objective, and their outputs are integrated through a structure-consistent assembly module, which constrains discrepancies.
> Second, our quantitative and qualitative results show that the generated shapes achieve state-of-the-art fidelity, diversity, and symmetry coherence, indicating that any potential accumulated errors are negligible in practice. We will clarify this design rationale in the revision.
>
> **References**
>
> Amin Mansour, E., Zheng, H., & Katzschmann, R. K. (2024, February). Fast point cloud to mesh reconstruction for deformable object tracking. In International Conference on Robotics, Computer Vision and Intelligent Systems (pp. 391-409). Cham: Springer Nature Switzerland.
>
> Hanocka, R., Metzer, G., Giryes, R., & Cohen-Or, D. (2020). Point2mesh: A self-prior for deformable meshes. arXiv preprint arXiv:2005.11084.

---

> > ### Comment · Reviewer_E7K3 · 2025-11-27
> >
> > I appreciate the detailed rebuttal. Overall, the proposed method is solid. However, relying on low-resolution point clouds as the 3D representation diminishes the practical impact, especially in light of recent advances in high-fidelity 3D generation. Therefore, I will maintain my current score.

---

> ### Author Response · Authors · 2025-11-28
>
> Thank you for the positive assessment. Regarding resolution, while point clouds remain a standard and widely used representation in generative 3D modeling, they can be converted into meshes via surface reconstruction methods. Higher resolution/fidelity is also achievable by training on denser point clouds, applying point cloud upsampling, or integrating modern surface reconstruction modules.
>
> Our primary contribution is a structure-aware generative framework that explicitly models parts and symmetry, a capability not addressed in prior work. While our current implementation uses standard-resolution point clouds for fair comparison with prior baselines, the proposed architecture can be trained directly on higher-resolution point sets if needed.
>
> Once again, we appreciate the reviewer's feedback and will expand the discussion to make these points clearer in the revised version.

---

### Official Review · Reviewer_6Uas · 2025-10-29

**Soundness:** 3
**Presentation:** 2
**Contribution:** 4
**Rating:** 8
**Confidence:** 3

**Summary:**

The paper proposes “Quartet of Diffusions,” a structure-aware point-cloud generative framework that factorizes generation into four diffusion models: a global shape latent, a symmetry group, distinct parts, and an assembly model. Each model is learned separately and connected via conditioning, which enforces part symmetry and yields realistic assemblies. The method is reported to be computationally efficient in training (despite four models) and outperforms prior work on standard metrics (e.g., Chamfer / 1-NNA) and on a newly introduced symmetry-awareness metric (SDI).

**Strengths:**

-	Clear conceptual decomposition: Factorizing generation into interpretable aspects (shape latent, symmetry, parts, assembly) is an elegant way to inject structure and inductive bias that matches human intuition about object design.
-	Solid empirical gains: The method shows strong improvements on classical generation metrics (1-NNA / Chamfer) and on the proposed symmetry metric (SDI), indicating both fidelity and structural plausibility.
-	Symmetry guarantees: Enforcing symmetry in the part generation produces more realistic, consistent results for symmetric objects.
-	Reasonable compute: The paper claims modest GPU/training cost despite four separate diffusion models, which makes the approach more practical than it may first appear.
-	Rigorous presentation: The paper is mathematically well-presented where needed.

**Weaknesses:**

-	While the authors introduced an evaluation metric for symmetry-awareness, the work does not explicitly evaluate the part-awareness such as in SeaLion (Zhu et al., 2025).
-	The method is only evaluated on three object classes of ShapeNet.
-	Related work could be improved and be made more readable. The section seems only like an assembly of many citations without having a flow content-wise.
-	The figures can be improved as often the point clouds overlap with text boxes, making the figures less readable and enjoyable (Fig. 1, 2, 3)

**Questions:**

-	By explicitly defining diffusion models for the four aspects (shape latent, symmetry, parts, assembly), the authors induce a bias into the model. Is it possible to investigate the impact of each of the four aspects individually?

---

> ### Author Response · Authors · 2025-11-20
>
> Thank you for recognizing the strengths and providing insightful and valuable comments on our work. Below we address each point in the Weaknesses and Questions sections.
>
> **Responses to Weaknesses**
>
> - **Part-awareness evaluation**: We sincerely thank the reviewer for pointing out the relevant work. To strengthen our evaluation, we have added quantitative comparisons with two recent and representative methods that explicitly leverage part annotations: **DiffFacto** (Nakayama et al., 2023) and **SeaLION** (Zhu et al., 2025). We report 1-NNA (p-CD) (Zhu et al., 2025) for part assembly quality and SDI for symmetry compliance.
>
> For airplane:
> | Method              | 1-NNA (p-CD) (↓)  | SDI (CD) (↓) |
> | :------------------ | :-------: | :-----: |
> | DiffFacto (Nakayama et al., 2023) |   81.7       |        252    |
> | SeaLION (Zhu et al., 2025)      |     65.4     |       132    |
> | Quartet (ours) |   **64.3**   |    **25.7**  |
>
> For car:
> | Method              | 1-NNA (p-CD) (↓)  | SDI (↓) |
> | :------------------ | :-------: | :-----: |
> | DiffFacto (Nakayama et al., 2023) |   90.5       |        278    |
> | SeaLION (Zhu et al., 2025)      |     73.1     |       215    |
> | Quartet (ours) |   **56.7**   |    **25.7**  |
>
> For chair:
> | Method              | 1-NNA (p-CD) (↓)  | SDI (↓) |
> | :------------------ | :-------: | :-----: |
> | DiffFacto (Nakayama et al., 2023) |   77.3       |        325    |
> | SeaLION (Zhu et al., 2025)      |    63.1     |       361    |
> | Quartet (ours) |   **53.9**   |    **28.9**  |
>
> Our method demonstrates **consistently superior performance** across all metrics, indicating strong part alignment and symmetry preservation. We will include this expanded comparison and table in the revised version.
>
> - **Evaluation on more classes**: Thank you for your suggestion. As **all prior methods** on 3D point cloud generation in Tab. 1 were evaluated on the three ShapeNet representative categories: airplanes, cars, and chairs, we have followed this standard practice. These categories are widely used due to their diverse part structures and well-annotated symmetry patterns, and they allow for direct comparison with all existing baselines. While extending the evaluation to additional categories or more complex datasets is feasible, it would reduce the pool of comparable methods due to limited prior results or require extensively retraining all baseline methods on new datasets. That said, we agree that broader evaluation would provide additional insights, and we are happy to include results on additional categories or datasets in the revised version.
>
> - **Improve related work**: Thank you for the suggestion. We will make related work more readable in our revision.
>
> - **Improve figures**: We thank the reviewer for this helpful suggestion. We agree that the figures can be made clearer. In the revised version, we will **adjust the layout and scaling of the point clouds** to avoid overlap with text boxes, and **refine the figure captions and spacing** to ensure better readability and visual balance.
>
> **Responses to Questions**
>
> - Thanks for pointing this out. We have indeed performed an **ablation study** in Sec. 4.4 to assess the contribution of each diffusion component individually. Our results show that all four diffusion models are indispensable members of the Quartet, each contributing to the generation of high-quality, diverse point clouds with strong symmetry.
>
> **References**
>
> Nakayama, G. K., Uy, M. A., Huang, J., Hu, S. M., Li, K., & Guibas, L. (2023). Difffacto: Controllable part-based 3d point cloud generation with cross diffusion. In Proceedings of the IEEE/CVF International Conference on Computer Vision (pp. 14257-14267).
>
> Zhu, D., Di, Y., Gavranovic, S., & Ilic, S. (2025). SeaLion: Semantic Part-Aware Latent Point Diffusion Models for 3D Generation. In Proceedings of the Computer Vision and Pattern Recognition Conference (pp. 11789-11798).

---

### Official Review · Reviewer_wWCJ · 2025-10-29

**Soundness:** 2
**Presentation:** 3
**Contribution:** 3
**Rating:** 4
**Confidence:** 5

**Summary:**

This work proposes a 3D point cloud generative model that considers part generation with symmetry. The model uses a four-stage diffusion process: it first generates a global shape latent, then symmetry groups (represented by reflection planes), followed by part geometry, and finally an assembler that composes the shapes. Experimental results show the framework achieves better performance than various point cloud generation baselines while using a lower training budget. The ablation study validates the effectiveness of the proposed method.

**Strengths:**

- The explicit consideration of symmetry in the 3D point cloud generation process is a novel exploration, and the evaluation, especially the SDI metric in Tables 1 and 3, clearly demonstrates the effectiveness of the proposed framework.
- Additionally, the proposed framework achieves state-of-the-art performance compared to existing point cloud generation methods on standard metrics (1-NNA in Table 1), showcasing the quality and diversity of generated shapes.
- A thorough ablation study in Table 3 validates the necessity of each introduced module.

**Weaknesses:**

Despite the strong performance shown, I still have some concerns regarding the evaluation setting.

1. Data Split Concerns. In Table 1, the evaluation protocol follows PointFlow (Yang et al., 2019), which uses the full category dataset for train/test splitting. Since this work uses ShapeNetPart, a subset of that data, it is unclear whether the train/test split remains consistent. If not, the test set may have leaked into the training set. Clarification is needed to ensure fair evaluation.
2. Comparison Baselines. The proposed framework is compared against many point cloud generation methods. However, none of them (including SALAD and SPAGHETTI) use part annotations for training their generative models. The proposed comparisons do not seem comprehensive. Instead, I would suggest comparing with methods that use part labels when training generative models. Some examples are provided below:
- StructureNet: Hierarchical Graph Networks for 3D Shape Generation. SIGGRAPH Asia 2019.
- SDM-NET: Deep Generative Network for Structured Deformable Mesh. SIGGRAPH Asia 2019.
- PQ-NET: A Generative Part Seq2Seq Network for 3D Shapes. CVPR 2020.
- DSG-Net: Learning Disentangled Structure and Geometry for 3D Shape Generation. TOG 2021.
- DiffFacto: Controllable Part-Based 3D Point Cloud Generation with Cross Diffusion. ICCV 2023.
- PASTA: Controllable Part-Aware Shape Generation with Autoregressive Transformers. Arxiv 2024.

I suggest making minor adjustments to the framework's claims:

- The paper claims the framework supports fine-grained manipulation, but provides no visual examples or quantitative evaluations. This claim should be softened to avoid confusion.
- Various existing works consider symmetry in 3D shape generation (particularly for feature extraction), especially StructureNet and related approaches. More discussion on this point is recommended.

**Questions:**

Overall, I am convinced by the novelty and strong performance of the proposed framework. However, I have concerns about the fairness of the evaluation protocol and the absence of important baselines. Therefore, I am currently leaning toward rejection but am open to raising my rating if the following questions are addressed:

1. Data Split Clarification: Could you please clarify the train/test split used in your experiments? Since you use ShapeNetPart (a subset of the full ShapeNet dataset) while the baseline evaluation protocol follows PointFlow (Yang et al., 2019) which uses the full category dataset, it is unclear whether your split maintains consistency with the original protocol. Specifically, could you confirm that there is no data leakage between your training and test sets, and provide details on how many samples were used for training vs. testing?
2. Comparison with Part-Aware Baselines: The current baselines (including SALAD and SPAGHETTI) do not use part annotations during training, which makes the comparison potentially unfair given that your method explicitly leverages part labels. Could you include comparisons with some part-aware generative methods such as StructureNet (SIGGRAPH Asia 2019), SDM-NET (SIGGRAPH Asia 2019), PQ-NET (CVPR 2020), DSG-Net (TOG 2021), DiffFacto (ICCV 2023), and PASTA (Arxiv 2024)? This would provide a more comprehensive evaluation of your method's advantages when part supervision is available.

---

> ### Author Response · Authors · 2025-11-20
>
> We thank the reviewer for the insightful comments and valuable questions. Below we address each point in the Weaknesses/Questions section:
>
> 1. The ShapeNetPart dataset is a semantically annotated subset of ShapeNetCore, containing the same underlying shapes with additional part labels (at least for the classes airplane, car, and chair). In our experiments, we **follow the PointFlow (Yang et al., 2019) train/test split** at the shape level. We first identify the shape IDs used in PointFlow’s split and then load their corresponding part annotations from ShapeNetPart. This ensures that the **training and test shapes are disjoint** and that no leakage occurs. We will explicitly clarify this procedure in the revised manuscript to avoid ambiguity.
>
> 2. We sincerely thank the reviewer for pointing out these relevant works. To strengthen our evaluation, we have added quantitative comparisons with two recent and representative methods that explicitly leverage part annotations: **DiffFacto** (Nakayama et al., 2023) and **SeaLION** (Zhu et al., 2025). We report 1-NNA (p-CD) (Zhu et al., 2025) for part assembly quality and SDI for symmetry compliance.
>
> For airplane:
> | Method              | 1-NNA (p-CD) (↓)  | SDI (CD) (↓) |
> | :------------------ | :-------: | :-----: |
> | DiffFacto (Nakayama et al., 2023) |   81.7       |        252    |
> | SeaLION (Zhu et al., 2025)      |     65.4     |       132    |
> | Quartet (ours) |   **64.3**   |    **25.7**  |
>
> For car:
> | Method              | 1-NNA (p-CD) (↓)  | SDI (↓) |
> | :------------------ | :-------: | :-----: |
> | DiffFacto (Nakayama et al., 2023) |   90.5       |        278    |
> | SeaLION (Zhu et al., 2025)      |     73.1     |       215    |
> | Quartet (ours) |   **56.7**   |    **25.7**  |
>
> For chair:
> | Method              | 1-NNA (p-CD) (↓)  | SDI (↓) |
> | :------------------ | :-------: | :-----: |
> | DiffFacto (Nakayama et al., 2023) |   77.3       |        325    |
> | SeaLION (Zhu et al., 2025)      |    63.1     |       361    |
> | Quartet (ours) |   **53.9**   |    **28.9**  |
>
> Our method demonstrates **consistently superior performance** across all metrics, indicating strong part alignment and symmetry preservation. We will include this expanded comparison and table in the revised version.
>
> Regarding PASTA (Li et al. 2024), while we briefly discussed PASTA in Sec. 2 as a contemporary part-aware shape generation method, we were not able to include it in our quantitative comparisons because 1. to the best of our knowledge, PASTA has not yet been published in a peer-reviewed venue; 2 at the time of writing, the official implementation of PASTA was not publicly available; 3. PASTA generates mesh representations, whereas our method and all baselines operate on point clouds; and 4. PASTA is trained on a different dataset (PartNet (Mo et al. 2019b)). These factors make a direct comparison challenging without substantial reimplementation and retraining on the point cloud dataset. We will make these distinctions explicit in the revised paper.
>
> - **Fine-grained manipulation**: our framework indeed supports targeted part-level control through the disentangled part diffusion process. As shown in Figs. 1 and 4, specific semantic parts are varied while the remaining structure remains fixed, demonstrating controlled and localized generation. Nevertheless, we agree that the current visual examples could be highlighted more explicitly. In the revised version, we will (1) make these manipulations clearer in the figure captions and (2) soften the wording of our claim to avoid overstating the result.
> - **Existing works considering symmetry**: Thank you for pointing this out. In the revised version we will include more discussion on the related works considering symmetry in 3D generation, including StructureNet (Mo et al. 2019a) and related approaches.
>
> **References**
>
> Nakayama, G. K., Uy, M. A., Huang, J., Hu, S. M., Li, K., & Guibas, L. (2023). Difffacto: Controllable part-based 3d point cloud generation with cross diffusion. In Proceedings of the IEEE/CVF International Conference on Computer Vision (pp. 14257-14267).
>
> Mo, K., Guerrero, P., Yi, L., Su, H., Wonka, P., Mitra, N., & Guibas, L. J. (2019a). Structurenet: Hierarchical graph networks for 3d shape generation. arXiv preprint arXiv:1908.00575.
>
> Mo, K., Zhu, S., Chang, A. X., Yi, L., Tripathi, S., Guibas, L. J., & Su, H. (2019b). Partnet: A large-scale benchmark for fine-grained and hierarchical part-level 3d object understanding. In Proceedings of the IEEE/CVF conference on computer vision and pattern recognition (pp. 909-918).
>
> Songlin Li, Despoina Paschalidou, and Leonidas Guibas. Pasta: Controllable part-aware shape generation with autoregressive transformers. arXiv preprint arXiv:2407.13677, 2024
>
> Zhu, D., Di, Y., Gavranovic, S., & Ilic, S. (2025). SeaLion: Semantic Part-Aware Latent Point Diffusion Models for 3D Generation. In Proceedings of the Computer Vision and Pattern Recognition Conference (pp. 11789-11798).

---

> > ### Comment · Reviewer_wWCJ · 2025-11-28
> >
> > Thanks for your responses. They have addressed my issue, and I will raise the score to 6 accordingly.

---

> > > ### Author Response · Authors · 2025-11-28
> > >
> > > We are glad to hear that and thank you very much for raising the score to 6!

---

### Official Review · Reviewer_rXcP · 2025-11-01

**Soundness:** 3
**Presentation:** 3
**Contribution:** 3
**Rating:** 4
**Confidence:** 3

**Summary:**

The paper proposed Quartet of Diffusion, a part-level, symmetry conditioned point cloud diffusion model that is trained on Shapenet dataset with segmentation labels. After sampling a global shape latent code, the paper then samples both the part level code and a symmetry group that the shape obeys. Afterward, a set of scales, translations, and rotations are sampled that are used to assemble the shapes together. The method shows superior generation metrics compared to other point cloud generation methods. It also supports part level generation which keeping the rest of the parts fixed

**Strengths:**

1. The paper proposed a novel symmetry conditioned point-cloud generation method. The symmetry diffusion formulation proposed in the paper seems novel and could potentially be a contribution to the shape generation field
2. The generation quality surpasses many strong baseline including LION and ShapeGF. This suggests the overall generation quality.

**Weaknesses:**

1. I found the application provided by the authors to be insufficient to justify the modeling of parts and symmetry. Part-level generation is not a novel task and could be done without modeling the symmetry (See DiffFacto). It would be great to see further application enabled by the structure-aware modeling.
2. The training requires part-segmentation labels as well as closed vocabulary part classes. This limits the generalization ability of the network to larger dataset such as Objaverse.
3. The generated outcome is a point-cloud, which to me is not clear what it would be useful for. Can subsequent surface reconstruction algorithm be used to recover a higher fidelity shape?

**Questions:**

1. Are the assembled shapes always valid? Are there intersections or detachment between parts? Some metrics that quantifies this would be helpful (See SeaLION or DiffFacto for example metrics)
2. Some shapes don’t have all the parts. How does the part diffusion model shapes with missing parts?

---

> ### Author Response · Authors · 2025-11-20
>
> We thank the reviewer for the insightful comments and valuable questions. Below we address each point in the Weaknesses and Questions sections:
>
> **Responses to Weaknesses**
>
> 1.
>
> We appreciate the reviewer’s reference to DiffFacto (Nakayama et al., 2023), which represents an important step towards part-aware generative modeling. Our goal is not to merely replicate part-level generation, but to **complement this line of work** by explicitly incorporating *symmetry priors* alongside *part composition* to achieve **structure-aware generation**. While DiffFacto models part relationships, our approach further integrates *symmetry distributions* into the generative process itself, enabling outputs that are not only part-consistent but also globally symmetric and structurally coherent. Comparing Trio Var. 2 and the full Quartet model in the ablation study in Tab. 3, we see that explicit symmetry modelling yields substantial gains in output quality, diversity, and symmetry fidelity.
>
> Our current experiments focus on unconditional shape generation to isolate and quantify the impact of these priors on *shape quality, diversity, and coherence*. We also provided a **targeted manipulation experiment**: As shown in Figs. 1 and 4, specific semantic parts are varied while the remaining structure remains fixed, demonstrating controlled and localized generation. Beyond this, our structure-aware modeling also provides a foundation for other potential downstream applications such as interactive 3D generation and scene composition, where preserving global structure and symmetry is critical. We will clarify this motivation and add examples of potential applications in the revised version.
>
> 2.
>
> The core contribution of our method is enabling part- and symmetry-aware 3D shape generation, which naturally requires access to part segmentation and symmetry labels during training. However, this requirement does not fundamentally limit the applicability of our approach. For open-world datasets like Objaverse, recent advances in unsupervised and weakly supervised 3D shape segmentation (e.g., Shu et al. 2016; 2019) can be used to estimate the necessary supervision. These tools make it feasible to apply our part- and symmetry-aware pipeline to any 3D shape dataset, including those without explicit part or symmetry annotations.
>
> Moreover, our framework is modular and adaptable: if part annotation is unnecessary, unavailable, or unreliable, we can merge the part and assembler diffusions into a unified full-shape generator. This variant corresponds to Duet Var. 2 in our ablation study (Sec. 4.4) and still achieves competitive 1-NNA scores and provides a practical way to extend our approach to more general 3D datasets without part labels. We will clarify this flexibility and discuss the application to open-world settings in the revised paper.
>
> 3.
>
> We respectfully note that **point clouds are a standard and widely adopted 3D representation** in generative shape modeling, reconstruction, and robotics.
> Generative methods in this area, including autoencoders, GANs, flows, and diffusion models, typically output point sets due to their simplicity, flexibility, and differentiability. Furthermore, **surface reconstruction algorithms** (e.g., Amin Mansour et al., 2024; Hanocka et al. 2020) can directly convert our generated point clouds into watertight meshes with high fidelity. We will further elaborate on this potential in the revised version.

---

> ### Author Response · Authors · 2025-11-20
>
> **Responses to Questions**
>
> 1.
>
> All assembled shapes we visualize are structurally valid and do not exhibit intersections or detachment. Although SeaLION (Zhu et al. 2025) is a contemporary work and was not published at the time of writing, to strengthen our evaluation, we have added quantitative comparisons using 1-NNA (p-CD) (Zhu et al., 2025) for part assembly quality and SDI for symmetry compliance. We compare against two recent and representative baselines that explicitly leverage part annotations: **DiffFacto** (Nakayama et al., 2023) and **SeaLION** (Zhu et al., 2025).
>
> For airplane:
> | Method              | 1-NNA (p-CD) (↓)  | SDI (CD) (↓) |
> | :------------------ | :-------: | :-----: |
> | DiffFacto (Nakayama et al., 2023) |   81.7       |        252    |
> | SeaLION (Zhu et al., 2025)      |     65.4     |       132    |
> | Quartet (ours) |   **64.3**   |    **25.7**  |
>
> For car:
> | Method              | 1-NNA (p-CD) (↓)  | SDI (↓) |
> | :------------------ | :-------: | :-----: |
> | DiffFacto (Nakayama et al., 2023) |   90.5       |        278    |
> | SeaLION (Zhu et al., 2025)      |     73.1     |       215    |
> | Quartet (ours) |   **56.7**   |    **25.7**  |
>
> For chair:
> | Method              | 1-NNA (p-CD) (↓)  | SDI (↓) |
> | :------------------ | :-------: | :-----: |
> | DiffFacto (Nakayama et al., 2023) |   77.3       |        325    |
> | SeaLION (Zhu et al., 2025)      |    63.1     |       361    |
> | Quartet (ours) |   **53.9**   |    **28.9**  |
>
> Our method demonstrates **consistently superior performance** across all metrics, indicating strong part alignment and symmetry preservation. We will include this expanded comparison and table in the revised version.
>
> 2.
>
> For shapes that do not contain all parts, our part diffusion model includes a **special “null-part” token** to explicitly represent missing components. This allows the model to handle variable part cardinalities without degradation in overall structure or quality. We will clarify this mechanism and add examples in the revision.
>
> **References**
>
> Amin Mansour, E., Zheng, H., & Katzschmann, R. K. (2024, February). Fast point cloud to mesh reconstruction for deformable object tracking. In International Conference on Robotics, Computer Vision and Intelligent Systems (pp. 391-409). Cham: Springer Nature Switzerland.
>
> Hanocka, R., Metzer, G., Giryes, R., & Cohen-Or, D. (2020). Point2mesh: A self-prior for deformable meshes. arXiv preprint arXiv:2005.11084.
>
> Nakayama, G. K., Uy, M. A., Huang, J., Hu, S. M., Li, K., & Guibas, L. (2023). Difffacto: Controllable part-based 3d point cloud generation with cross diffusion. In Proceedings of the IEEE/CVF International Conference on Computer Vision (pp. 14257-14267).
>
> Shu, Z., Qi, C., Xin, S., Hu, C., Wang, L., Zhang, Y., & Liu, L. (2016). Unsupervised 3D shape segmentation and co-segmentation via deep learning. Computer Aided Geometric Design, 43, 39-52.
>
> Shu, Z., Shen, X., Xin, S., Chang, Q., Feng, J., Kavan, L., & Liu, L. (2019). Scribble-based 3D shape segmentation via weakly-supervised learning. IEEE transactions on visualization and computer graphics, 26(8), 2671-2682.
>
> Zhu, D., Di, Y., Gavranovic, S., & Ilic, S. (2025). SeaLion: Semantic Part-Aware Latent Point Diffusion Models for 3D Generation. In Proceedings of the Computer Vision and Pattern Recognition Conference (pp. 11789-11798).

---

> ### Comment · Reviewer_rXcP · 2025-11-27
>
> I appreciate the thorough response by the authors, addressing my concerns. The quantitative comparison with SeaLION and DiffFacto clearly shows your methods' advantage especially when it comes to symmetry modeling.
>
> However, I am still not fully convinced that the authors explored the full potential of modeling an additional symmetry prior. The applications and interactions you indicated, such as targeted manipulation, structure-aware modeling, do not require the modeling of symmetry prior -- methods such as SeaLION and DiffFacto were able to achieve these already. So what application does modeling symmetry enable besides better reconstruction quality?
>
> Further, I agree with Reviewer E7K3's point that the generated points are too low resolution for surface reconstruction algorithms. So I'm not sure if your method's output can be directly converted to surfaces without additional method development.
>
> Given the updated results, I will raise my score to a 6. However, I still have lingering concerns regarding the applications showcased in the paper as well as the validity of surface reconstruction from generated points.

---

> ### Author Response · Authors · 2025-11-28
>
> We sincerely thank the reviewer for the positive update, for raising the score, and for raising these helpful points.
>
> **On the utility of symmetry.**
>
> Our motivation for modeling explicit symmetry distributions is to enable
> - physical correctness,
> - guaranteed global symmetry, even under large part variations, and
> - improved robustness in assembly, as shown by the gains over Trio variants in Table 3 and over quantitative and qualitative evaluations.
>
> We agree that additional applications could highlight this more explicitly, and we will clarify this in the revision.
>
> **On surface reconstruction from generated points.**
>
> We appreciate this concern. While point clouds remain a standard and widely used representation in generative 3D modeling, they can be converted into meshes via surface reconstruction methods. Higher resolution/fidelity is also achievable by training on denser point clouds, applying point cloud upsampling, or integrating modern surface reconstruction modules.
>
> Our primary contribution is a structure-aware generative framework that explicitly models parts and symmetry, a capability not addressed in prior work. While our current implementation uses standard-resolution point clouds for fair comparison with prior baselines, the proposed architecture can be trained directly on higher-resolution point sets if needed.
>
> Once again, we are grateful for the reviewer's constructive feedback and will incorporate these clarifications to strengthen the presentation of both the motivation and practical applications of our method.

---

### Author Response · Authors · 2025-11-29
**Summary of rebuttal**

We appreciate the reviewers' positive assessment of our contribution. The initial ratings were (8, 6, 4, 4). We are pleased to see that the reviewers appreciated the novelty and contribution of our work to propose a structure-aware generative framework that explicitly models parts and symmetry, a capability not addressed in prior work. We are also happy to see that the reviewers enjoyed our clear presentation and thorough and rigorous experiments showing SOTA results.

During the rebuttal, we clarified key points, addressed concerns raised by the reviewers, and conducted additional experiments, including comparisons with more recent part-aware baselines using part-aware metrics. We sincerely thank the reviewers for the valuable feedback that helped strengthen the paper.

Due to the OpenReview incident, we were unable to continue discussion with or to receive the feedback from all reviewers, but before the incident, both reviewers who initially gave a 4 increased their scores to 6. Thus, the post-rebuttal **ratings are at least (8, 6, 6, 6)**.

---

### Meta-Review · Area_Chair_gXCU · 2026-01-06

**Summary:**

The paper received mixed reviewers initially. The authors answered the questions very well. Due to some existing issues (e.g., applications of point clouds, limited datasets), I would recommend a poster paper.

**Reviewer Concerns:**

The authors addressed the following concerns very well:
1. Symmetry modeling
2. Unsupervised learning
3. Evaluation protocol
4. Efficiency

**Reviewer Scores:**

Based on my experience, reviewers would tend to accept the paper.

---

### Decision · Program_Chairs · 2026-01-26

Accept (Poster)